# An ON-type direction-selective ganglion cell in primate retina

Anna Y. M. Wang[1,2], Manoj M. Kulkarni[1,2], Amanda J. McLaughlin[1,2], Jacqueline Gayet[1,2], Benjamin E. Smith[1,3], Max Hauptschein[1], Cyrus F. McHugh[1,3], Yvette Y. Yao[1] & Teresa Puthussery[1,2✉]

To maintain a stable and clear image of the world, our eyes reflexively follow the direction in which a visual scene is moving. Such gaze-stabilization mechanisms reduce image blur as we move in the environment. In non-primate mammals, this behaviour is initiated by retinal output neurons called ON-type direction-selective ganglion cells (ON-DSGCs), which detect the direction of image motion and transmit signals to brainstem nuclei that drive compensatory eye movements[1]. However, ON-DSGCs have not yet been identified in the retina of primates, raising the possibility that this reflex is mediated by cortical visual areas. Here we mined single-cell RNA transcriptomic data from primate retina to identify a candidate ON-DSGC. We then combined two-photon calcium imaging, molecular identification and morphological analysis to reveal a population of ON-DSGCs in the macaque retina. The morphology, molecular signature and GABA (γ-aminobutyric acid)-dependent mechanisms that underlie direction selectivity in primate ON-DSGCs are highly conserved with those in other mammals. We further identify a candidate ON-DSGC in human retina. The presence of ON-DSGCs in primates highlights the need to examine the contribution of subcortical retinal mechanisms to normal and aberrant gaze stabilization in the developing and mature visual system.

Visual systems in animals ranging from flies to humans have evolved sophisticated gaze-stabilization reflexes that maintain a stable image of the world on the photoreceptor array during self motion. One such mechanism, the optokinetic reflex (OKR), generates eye movements that track the velocity of global motion of the visual field, thereby stabilizing the retinal image. In lower vertebrates and mammals, retinal ON-DSGCs provide the crucial directional input to the accessory optic system (AOS), the collection of midbrain nuclei that drive the OKR[2–4]. ON-DSGCs respond strongly to motion in their preferred direction and weakly to motion in the opposite, null direction[1,5]. Despite extensive characterization in other species, it remains unknown whether ON-DSGCs are present in the primate retina and thus whether directional signals that drive the OKR are generated exclusively at the level of the visual cortex[6].

There are indications that ON-DSGCs might be present in primates. Similar to lower vertebrates, tracer studies have shown retinal ganglion cell (RGC) projections to the AOS[7–10], and a morphological candidate for a primate ON-DSGC has been described recently[8]. However, ON-DSGCs have not been functionally identified in primates, possibly owing to the difficulty of sampling relatively rare cell types, even using approaches that sample from thousands of RGCs simultaneously[11,12]. Indeed, many of the dozen or more uncharacterized primate RGC types each make up only a small fraction (around 1%) of the total RGC population[12–14]. The preponderance of midget and parasol ganglion cells underlies the general belief that primates do the majority of computation in the cortex, implicitly discounting the importance of uncharacterized retinal inputs[15]. Here we use a novel approach to determine the functional, molecular and morphological properties of an ON-DSGC in the primate retina. We show that these cells are similar to ON-DSGCs in other mammals, consistent with a conserved functional role.

## A candidate ON-DSGC in primate retina

We mined an existing single-cell transcriptomic dataset from macaque (*Macaca fascicularis*) retina to identify a candidate ON-DSGC (Gene Expression Omnibus (GEO) accession GSE118480[14]). Our goal was to identify RGC types that contain the α2 subunit of the type A GABA receptor[16,17] (GABA_A receptor) (encoded by *GABRA2*) and glycine receptor alpha subunits (encoded by *GLRA* genes), both of which are components of the ON-DSGC circuit in other mammals[18–20]. Our search for a candidate was further aided by the fact that in the primate retina, the majority of RGC types express markedly lower levels of inhibitory receptors in the fovea compared with the periphery[14,21]. If ON-DSGCs are present in the foveal retina, then they should be distinguished from other RGC types by their expression of these GABA_A and glycine receptors, which are required for their distinct functional properties. Indeed, two RGC types of unknown function showed comparable expression of these receptors in foveal and peripheral retina: peripheral (p)RGC10 and pRGC16 (Fig. 1a). We pursued pRGC10 as our primary candidate for an ON-DSGC owing to its higher expression of *GABRA2* and *GLRA2* and because it seems to be the transcriptomic orthologue of the AOS-projecting ON-DSGCs in mouse[14,22–25].

[1]Herbert Wertheim School of Optometry and Vision Science, Berkeley, CA, USA. [2]Helen Wills Neuroscience Institute, Berkeley, CA, USA. [3]Vision Science Graduate Program, University of California, Berkeley, Berkeley, CA, USA. ✉e-mail: tputhussery@berkeley.edu

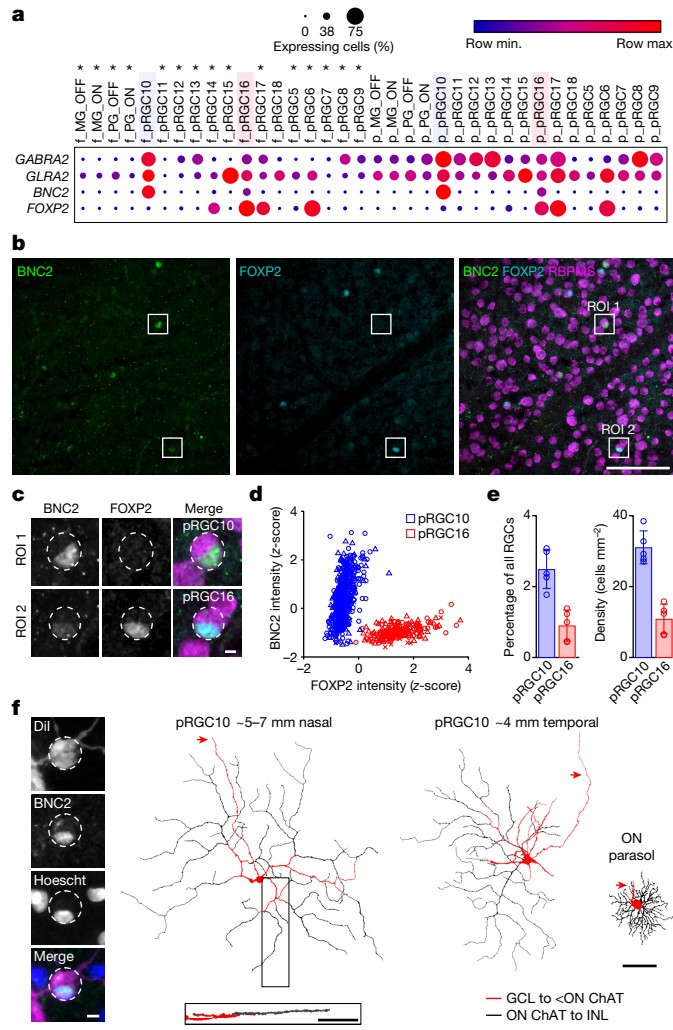

**Fig. 1 | BNC2 is a putative marker of macaque ON-DSGCs. a**, Dot plots showing relative expression of inhibitory receptors and cell markers in corresponding foveal (f_) and peripheral (p_) macaque RGC transcriptomic clusters. Circle size indicates the percentage of cells expressing the gene and colour indicates the relative transcript count in expressing cells. Raw data from GEO accession GSE118480[14]. Asterisks denote cell types where the difference between foveal and peripheral *GABRA2* and/or *GLRA2* expression is more than 2 log fold and $P < 0.05$ by two-sided Wilcoxon rank-sum test. Max., maximum; min., minimum. Blue and pink highlighting indicates the pRGC10 and pRGC16 clusters, respectively. **b**, Representative confocal image of macaque GCL immunolabelled for BNC2, FOXP2 and RBPMS. **c**, Enlargements of indicated regions of interest (ROIs) in **b**. Images in **b** and **c** were 2D median filtered before maximum *z*-projection. **d**, *z*-Score-normalized intensities of BNC2 and FOXP2 for each BNC2+ RGC (*n* = 733 cells, 3 retinas). Each point represents a single cell, and different symbol shapes indicate data from different macaques. Cells were clustered using *k*-means analysis. **e**, Summary of percentages and cell density for pRGC10, with pRGC16 shown for comparison (50,050 RGCs from *n* = 5 retinas, 3.5–8.5 mm nasal equivalent eccentricity). Data are mean ± s.d. **f**, Left, confocal image of the soma of a DiI-filled cell showing BNC2 staining in its nucleus (Hoescht). Top right, morphological reconstructions of DiI-filled BNC2+ cells pseudocoloured by stratification depth. Approximate equivalent eccentricities are indicated above the images. Bottom right, side projection of the boxed area of the left cell. Primary dendrites extend in the GCL before diving into the inner plexiform layer (IPL) towards the inner nuclear layer (INL). An ON-parasol ganglion cell at ~4 mm temporal eccentricity is shown on the far right for comparison. Red arrows indicate axons. Scale bars, 100 µm (**b**); 5 µm (**c**); 50 µm (**f**, left); 100 µm (**f**, top and bottom right).

Transcripts encoding the zinc-finger family transcription factor basonuclin 2 (*BNC2*) are enriched in pRGC10 cells[14], but *BNC2* is also expressed at lower levels in pRGC16 cells. To distinguish these cell types, we identified a second gene product—*FOXP2*, which encodes the transcription factor forkhead box P2—that is expressed in pRGC16 but not pRGC10[14] (Fig. 1a). Consistent with transcript expression patterns, double labelling with antibodies against BNC2 and FOXP2 labelled a sparse subset of RGCs (Fig. 1b,c), which could be reliably clustered into pRGC10 and pRGC16 types on the basis of the relative expression of the two markers (Fig. 1d). Of all of the BNC2+ RGCs, pRGC10 constituted the majority at 74.5 ± 6.8% (1,157 out of 1,584 cells, *n* = 5 retinas), similar to the percentage predicted from the transcriptome (68% pRGC10 and 32% pRGC16; GEO accession GSE118852[14]). pRGC10 cells were found only in the ganglion cell layer (GCL) and comprised 2.5 ± 0.54% of all RGCs—these cells had a density of 31.1 ± 4.7 cells per mm² in peripheral nasal macaque retina (*n* = 5 retinas; Fig. 1e).

A single cell type that evenly covers or 'tiles' the retina should have somas positioned in a relatively regular mosaic[26]. The spacing of pRGC10 soma positions was inconsistent with such regularity (Extended Data Fig. 1). A potential explanation is that primates are similar to other mammals, which have at least three distinct populations of ON-DSGCs, each of which encodes different preferred directions and tiles the retina in an independent regular mosaic[27–29]. Consistent with this idea, analysis of the spatial positioning of pRGC10 somata suggested the presence of multiple mosaics (Extended Data Fig. 1f). Moreover, closer examination of the pRGC10 transcriptomic cluster described by Peng et al.[14] showed three putative subclusters within the pRGC10 group (Extended Data Fig. 2a). In the mouse retina, ON-DSGCs that respond to upward motion in the visual field can be distinguished by expression of the gene *FSTL4*[3] (also known as *SPIG1*). Working on the assumption that the same might be true in primates, we compared the expression of *FSTL4* between the putative pRGC10 transcriptomic subclusters (Extended Data Fig. 2b,c). In line with our expectation, *BNC2* showed similar expression across subclusters, whereas *FSTL4* expression was restricted to a single subcluster. Fluorescence in situ hybridization of the macaque GCL, confirmed this expression pattern, with *FSTL4* being present in a subset of *BNC2*+ RGCs (Extended Data Fig. 2d,e). These data suggest that the pRGC10 cluster is composed of at least three subtypes.

Prior studies using retrograde labelling of RGCs from injections into the macaque nucleus of the optic tract-dorsal terminal nucleus (NOT-DTN), the central target of horizontal-preferring DSGCs, revealed a higher density of RGCs along the horizontal midline of the retina[7]. These findings prompted us to examine the wider spatial distribution of the candidate BNC2+ ON-DSGCs. Indeed, BNC2+ RGCs in the nasal half of a single macaque retina were asymmetrically distributed, with the highest density on the horizontal midline (approximately 43 cells per mm²) and a relatively flat density gradient between the optic disc and peripheral nasal retina (Extended Data Fig. 3). Notably, both the density and the proportion of BNC2+ RGCs were higher in superior retina compared with inferior retina, which aligns with reported asymmetries in OKR gain in the upper visual field compared with the lower visual field[30]. These density maps guided our choice of recording area in the functional experiments that follow.

To determine whether the morphology of pRGC10 cells was similar to ON-DSGCs in other species, we pre-labelled fixed retinas with antibodies against BNC2 and then filled BNC2+ cells with the lipophilic tracer DiI (Fig. 1f, left). The filled cells had wide-field monostratified dendritic arbours (Fig. 1f, top right), with thick primary dendrites extending in the GCL before giving rise to thinner dendrites in the ON-sublamina of the IPL (Fig. 1f, bottom). The dendrites branched with few crossings and had some 'recursive' dendritic segments that reflected back towards the soma (Fig. 1f; for additional examples see Supplementary Videos 1 and 2), similar to ON-DSGCs in other mammals[19,31] and to the recursive monostratified RGCs described previously in macaque and marmoset

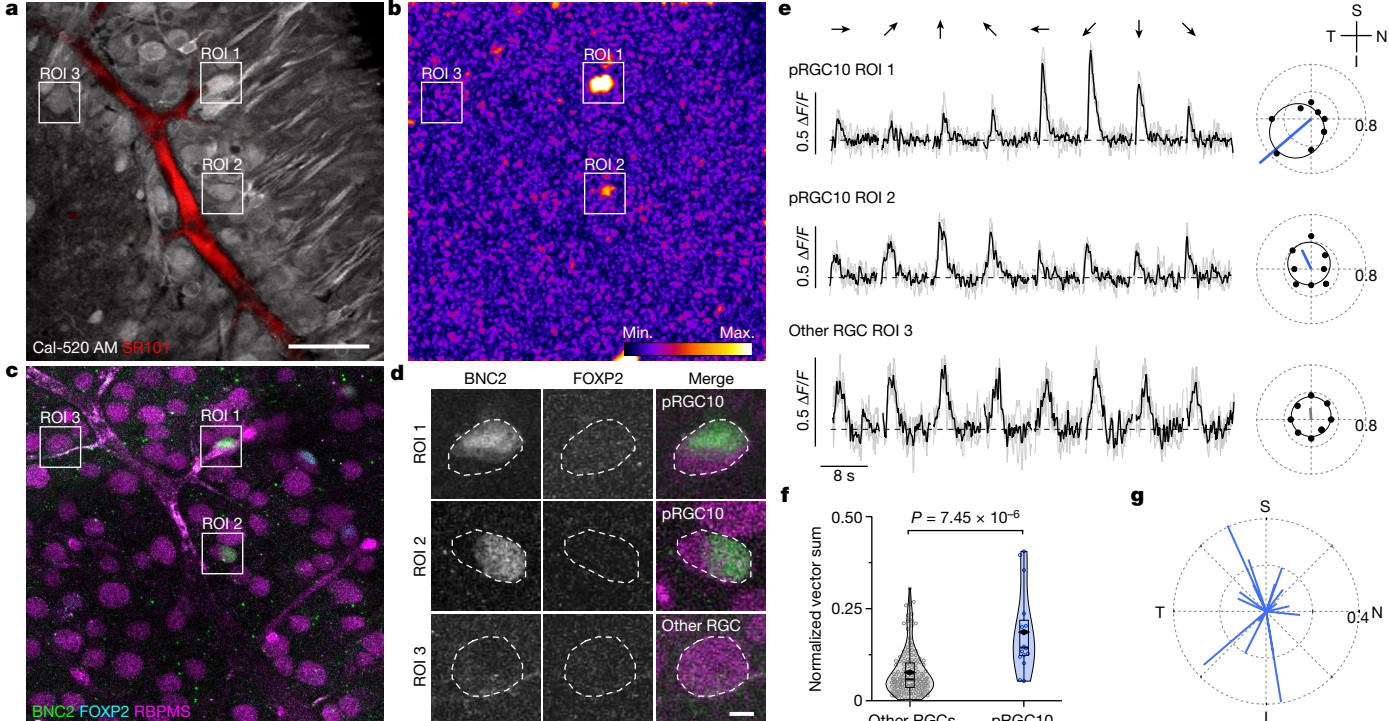

**Fig. 2 | pRGC10 cells are direction-selective. a**, Whole-mounted macaque retina loaded with Cal-520 AM and sulforhodamine 101 (SR101)—a vascular marker—in the GCL and imaged with a two-photon microscope (standard deviation *z*-projection). **b**, Vector sum mapping of the same field highlighting pixels with directional responses. **c**, Post hoc immunostaining of experimental sample from **a**,**b** for BNC2, FOXP2 and RBPMS. **d**, ROIs from **c**, showing examples of pRGC10 cells (BNC2⁺FOXP2⁻RBPMS⁺, ROI 1 and ROI 2) and another RGC type (BNC⁻FOXP2⁻RBPMS⁺, ROI 3). Images in **c**,**d** are average *z*-projections; cells in **d** are averaged over a smaller *z* depth for better visualization. **e**, Left, Δ*F*/*F* calcium responses of cells in ROI 1, 2 and 3 to bars (500 μm wide) drifting in the 8 directions shown by arrows above traces (0–360°, 45° increments). Responses are averages (black line) of three trials (grey lines) for each stimulus angle from two different

scans. The dashed line indicates pre-stimulus baseline. Right, corresponding polar plots showing peak Δ*F*/*F* as a function of angle. Von Mises fit (solid black line) and non-normalized vector sum (blue and grey lines) for traces on the left. The orientation of the retina is indicated at top right: S, superior; N, nasal; I, inferior; T, temporal. **f**, Violin and box plots showing normalized vector sum values for pRGC10 (*n* = 16 cells) and other RGCs (*n* = 140 cells) from 6 retinas. The boxes cover the interquartile range, whiskers show minimum and maximum values, the solid black line shows the median, and the diamond represents the mean. Two-sided Wilcoxon rank-sum test. **g**, Polar plots showing normalized vector sums for all recorded pRGC10 cells. Scale bar, 50 μm (**a** also applies to **b**,**c**), 5 μm (**d**).

retina[8,13]. The filled cells had an average dendritic field diameter of 388 ± 82 μm (*n* = 7 cells) and dendritic area of 119,892 ± 48,721 μm² (*n* = 7 cells, nasal equivalent eccentricity of 4–7 mm). Together with the cell density of pRGC10 cells (Fig. 1e), we calculated a coverage factor of approximately 3.5, similar to the coverage factor of recursive monostratified RGCs recently reported in the macaque retina[13]. A coverage factor greater than one is indicative of extensive overlap of dendritic arbours of neighbouring cells and is in line with the mosaic and molecular subtype analyses described above, which suggest the presence of multiple pRGC10 subtypes. Together, these results demonstrate that pRGC10 cells probably comprise multiple subtypes and have the expected stratification and morphology of ON-type DSGCs.

### Functional identification of ON-DSGCs

To determine whether the putative ON-DSGCs were direction-selective, we used two-photon calcium imaging to record responses of neurons in the macaque GCL to bars drifting in eight different directions at speeds known to activate ON-DSGCs in other mammals[2,19] (Fig. 2 and Methods). Candidate ON-DSGCs were identified by calculating a vector sum (Methods) for each pixel in the scan area. Clusters of highly directional pixels revealed the locations of the ON-DSGC somas, whose identity as pRGC10 cells was confirmed post hoc by immunolabelling for BNC2, FOXP2 and RBPMS (Fig. 2a–d). In line with our prediction, pRGC10 cells exhibited ON responses that were directionally tuned (Fig. 2e; see also Extended Data Fig. 4 for spot responses). On average, the directional tuning (normalized vector sum; Methods) was

significantly higher for pRGC10 cells compared with other responsive RGCs in the same scan fields (Fig. 2f; pRGC10, 0.14 (0.12–0.22) (median (interquartile range)), *n* = 16 cells; versus other RGCs, 0.06 (0.04–0.10), *n* = 140 cells; *P* = 7.45 × 10⁻⁶, Wilcoxon rank-sum test, 6 retinas). The mosaic and molecular analysis above predicts the presence of three populations of cells defined by their preferred directions; however, given the relatively small number of recovered pRGC10 cells, we were unable to resolve such a distribution (Fig. 2g). Some RGCs that did not belong to the pRGC10 group also showed higher normalized vector sums; the molecular identity of these cells is not yet known but they could include the ON–OFF DSGCs described recently in macaque[13] or a transient ON-DSGC type that has been described in rabbit[32,33] and mouse[22]. Another characteristic feature of ON-DSGCs in other mammals is their slow velocity tuning[18–20]. We thus tested a subset of identified ON-DSGCs to bars moving in the preferred and null directions at speeds ranging from 125–2,000 μm s⁻¹ (0.57–9 degrees per second; Extended Data Fig. 4d). Although cells responded over a broad range of stimulus velocities, maximal responses were obtained around 1 degree per second, consistent with the speed tuning of ON-DSGCs in other mammals[19,20]. Finally, since the *Bnc2* transcript is also enriched in mouse ON-DSGCs (mouse transcriptomic cluster Novel_10 (refs. 22,23)), we validated our overall approach by performing calcium imaging on mouse RGCs (Extended Data Fig. 5). Consistent with the macaque data, we identified BNC2⁺ ON-DSGCs in the mouse that clustered into the expected preferred directions and showed similar directional tuning to previous reports[2,34,35].

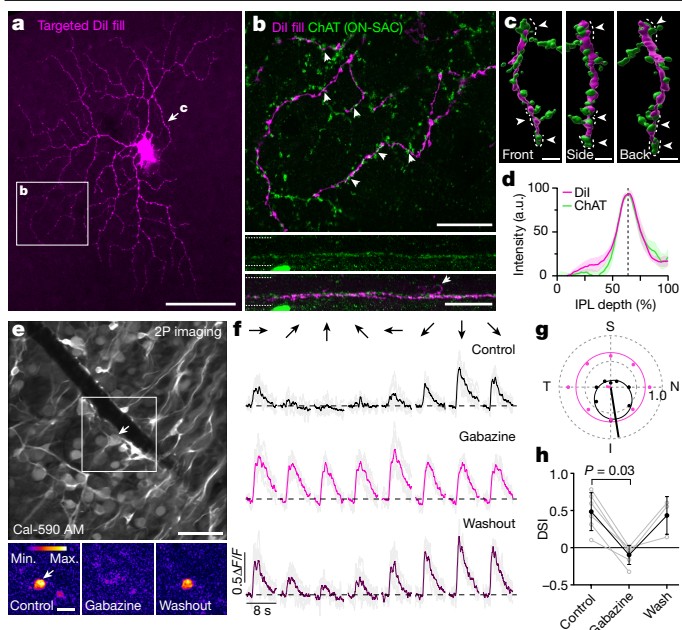

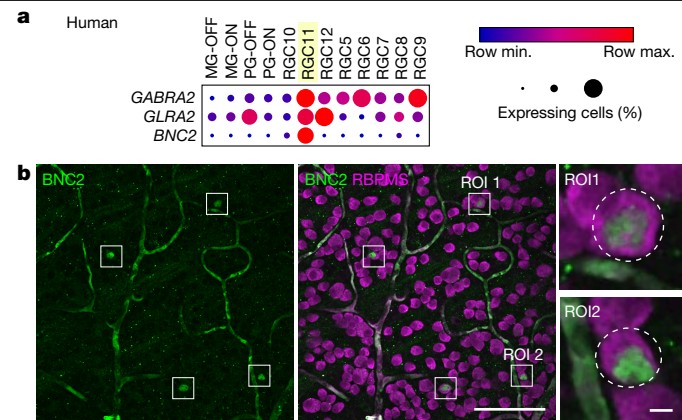

**Fig. 3 | GABAergic inhibition is required for directional tuning of primate ON-DSGCs. a,** Confocal projection of a DiI-filled pRGC10 cell. Regions of the dendritic arbour are shown in **b,c. b,** Top, ON-SAC dendrites showing putative contacts with DiI-filled cell (arrowheads). Maximum z-projection of 2.47 μm. **b,** Middle and bottom, side projections from indicated region in top image, showing ON-SAC dendrites alone (middle) and merged with the DiI channel (bottom). Note the ON-DSGC dendrite extending to the OFF-sublamina (arrowhead). Dotted lines delineate IPL borders. **c,** 3D surface renders show ON-SAC dendrites wrapped around (arrowheads and dotted lines) an ON-DSGC dendrite (indicated by arrow in **a**). **d,** Normalized intensity profiles of ChAT and DiI (mean ± s.d. (shaded region), n = 5 cells) with peaks at approximately 64% depth (line). a.u., arbitrary units. **e,** Top, example calcium imaging field (average z-projection) of the macaque GCL with an ON-DSGC indicated by arrow (see Extended Data Fig. 6 for immunostaining). Bottom, vector sum heat map of the ROI (top) before, during and after washout of gabazine (10 μM). **f,** Calcium responses of the ON-DSGC in **e**. Averages of six trials (bold) are overlaid on individual responses (grey). The dashed line shows the pre-stimulus baseline. Stimulus was as in Fig. 2e. **g,** Polar graph showing peak responses for the cell in **f**. Vector sum (straight lines) and von Mises fits (smooth lines) are shown for control (black) and gabazine (magenta). **h,** Average DSIs (black circles) for ON-DSGCs treated with gabazine (6 cells, 3 macaques), washout was obtained in 3 cells. Two-sided Wilcoxon signed rank test. Data are mean ± s.d.; grey circles show individual cells. Scale bars: 100 μm (**a**), 20 μm (**b**), 5 μm (**c**), 50 μm (**e**, top), 20 μm (**e**, bottom).

## Inhibitory mechanisms in ON-DSGCs

We next tested whether the circuit mechanisms underlying direction selectivity in primates are similar to those described in other species. In mice and rabbits, asymmetric GABAergic inhibition from ON-type starburst amacrine cells (SACs) is the major source of directional tuning of ON-DSGCs[36,37]. Thus, we first assessed whether the primate ON-DSGCs co-stratified with ON-SACs, which were labelled with antibodies for choline acetyltransferase (ChAT) (Fig. 3a,b). The dendrites of the DiI-filled ON-DSGCs showed extensive co-fasciculation with ON-SACs (Fig. 3b). Moreover, volumetric reconstructions revealed SAC dendrites wrapping around the dendrites of the ON-DSGCs (Fig. 3c and Supplementary Videos 3–5), similar to the 'wrap-around' synapses that have been described in ON-DSGCs in other species[38,39] and in a candidate ON-DSGC in primates[8]. Both ON-SACs and ON-DSGCs co-stratified at around 64% of the IPL depth (where 0% is the inner nuclear layer border and 100% is the GCL border) (Fig. 3d; n = 5 cells) similar to the recursive monostratified RGC in macaque retina[8] (note,

**Fig. 4 | A candidate ON-DSGC in human retina. a,** Dot plots showing the relative expression of genes encoding inhibitory receptor subunits and BNC2 in human RGCs. Circle size corresponds to the percentage of cells expressing the gene and colour indicates relative transcript count in expressing cells. Yellow highlighting indicates the RGC11 cluster. Raw data from GEO GSE148077[41]. **b,** Human GCL labelled for BNC2 and RBPMS with example RGC11 cells in ROIs. ROI 1 and ROI 2 are shown enlarged on the right. Images in **b** were 2D median filtered before maximal z-projection of the GCL. Scale bars: main image, 100 μm; enlarged ROI, 5 μm.

however, the reported stratification of recursive monostratified cells at around 80% IPL depth[13]). The surface contact area between the DiI and ON-SAC staining was significantly higher with the ChAT channel in the original orientation compared with when it was rotated 90° about the z-axis, suggesting synaptic connectivity between ON-SACs and the ON-DSGCs (normal, 1.86 ± 0.97% to rotated 0.51 ± 0.27%; P = 0.016, n = 7 cells). We occasionally found evidence of dendrites that extended vertically from the ON sublamina into the OFF sublamina of the IPL, where they terminated abruptly without branching into a second tier of dendrites (Fig. 3b, bottom and Supplementary Video 6). Such dendritic processes have been described in serial electron microscopy reconstructions of recursive monostratified RGCs[8], and were shown to receive OFF-bipolar cell input, consistent with a small OFF component seen in ON-DSGC calcium responses to spots of light (Extended Data Fig. 4a).

The results above are consistent with the ON-DSGCs receiving synaptic input from ON-SACs. Blocking GABAergic inhibition in other mammals eliminates SAC inputs and abolishes direction selectivity. Therefore, we used calcium imaging to test whether GABAergic inputs are required for directional tuning of primate ON-DSGCs. As before, we used a drifting bar stimulus and looked for directionally tuned pixels in the scan field to locate candidate ON-DSGCs. We then applied the GABA$_A$ receptor blocker gabazine (10–20 μM) in the bath solution, to test whether the directional tuning could be suppressed. Consistent with our anatomical data, gabazine reversibly abolished the directional tuning of the ON-DSGCs (Fig. 3e–h, directional selectivity index (DSI): control, 0.49 ± 0.25; gabazine, −0.10 ± 0.13; P = 0.03, n = 6 cells, Wilcoxon signed rank test). These cells were confirmed to be pRGC10 by post hoc immunostaining (Extended Data Fig. 6). Together, these results indicate that GABAergic inhibition from SACs is essential for directional selectivity in primate ON-DSGCs, suggesting that the fundamental circuit properties that generate direction selectivity are conserved across species[36,37,40].

## A putative ON-DSGC in human retina

Finally, we sought evidence for the presence of ON-DSGCs in the human retina. Prior work suggests that the transcriptomic orthologue of macaque pRGC10 is the human type RGC11[41]. Similar to the macaque ON-DSGC, the human orthologue expresses high levels of *GABRA2* and

*GLRA2* and can be distinguished from other human RGC types by high expression of *BNC2* (Fig. 4a; raw data from GEO accession GSE148077[41]). In line with the transcriptomic data, BNC2 antibodies labelled a sparse subset of human RGCs that made up 0.88% of the total RGC population (*n* = 2 retinas: retina 1, 75 out of 8,415 cells; retina 2, 54 out of 6,127 cells) and had an average density of 9.4 cells per mm$^2$ in peripheral nasal retina (Fig. 4b; retina 1, 5.6 cells per mm$^2$, retina 2, 13.14 cells per mm$^2$). As in the macaque retina, mosaic analysis suggested the presence of more than one mosaic (Voronoi domain regularity index: retina 1, 2.03 real and 2.04 simulated; retina 2, 2.34 real and 2.03 simulated (data not shown)). These data suggest that there is a functionally homologous ON-DSGC in the human retina.

## Discussion

Here we demonstrate the presence an ON-type DSGC in the primate retina. We show that these cells belong to a distinct transcriptional group and have morphology and inhibitory circuit connectivity consistent with ON-DSGCs that project to the AOS in other species[2,3]. Our study demonstrates the power of combining two-photon calcium imaging with molecular classification to identify rare RGC types, since the ON-DSGCs comprise only 1–2.5% of all primate RGCs. By linking ON-DSGC function to a transcriptomically defined cell type, we deduce that these cells are one of the few RGC types that show high molecular homology between mouse and primates[14,22,23,41]. Such high cross-species homology suggests that ON-DSGCs serve a critical and conserved ethological function.

The brainstem nuclei that drive the OKR in primates receive direct input from the retina[7–10] as well as from the medial temporal and medial superior temporal areas of the cortex[42]. We propose that the retinal input arises from the ON-DSGCs described here. The relative contribution of cortical and retinal inputs to the OKR remains unclear. During early human development, the monocular OKR shows a naso-temporal asymmetry[43] similar to that seen in afoveate animals such as rabbits whose OKR is driven by a reflexive subcortical mechanism[44]. The OKR becomes symmetric at 4 to 5 months of age, coinciding with the establishment of binocular vision and maturation of cortical input to the NOT-DTN in primates[43]. Thus, it has been proposed that the early OKR is driven primarily by retinal input, whereas cortical inputs dominate later in development[45]. However, a reflexive OKR can be evoked with certain visual stimuli in adults and after occipital lobectomy in primates, suggesting that the direct retinal pathway continues to have a role in gaze stabilization in the mature visual system[46,47]. Overall, our findings highlight the need for further studies to determine the relative contributions of direct retinal and cortical mechanisms to the human OKR.

Deficits in the subcortical OKR pathway may also have a role in human gaze-stabilization disorders. Nystagmus is characterized by involuntary, rhythmic eye movements that can lead to reduced visual acuity and unsteady vision[48] (oscillopsia). Some forms of infantile nystagmus are thought to arise from alterations in DSGC circuits[34,49]. For example, mutations in *FRMD7*—a gene implicated in infantile nystagmus and expressed in SACs—leads to loss of horizontal OKR in humans. The OKR phenotype is recapitulated in *FRMD7* mutant mice, which lack horizontally tuned DSGCs[34] owing to altered SAC inputs. Similarly, a form of nystagmus associated with congenital stationary night blindness has been proposed to arise from aberrant ON-DSGC activity[49]. The confirmed presence of ON-DSGCs in primates further supports the notion that some forms of nystagmus could arise from altered DSGC signalling[50].

More generally, our findings highlight the importance of multimodal cell classification for resolving the functions of primate RGCs of unknown function. Indeed, apart from the well-characterized midget, parasol and intrinsically photosensitive RGCs, the morphologies and functions of other transcriptomically defined primate RGC types are not yet known[14]. Our study illustrates a robust approach to characterize these remaining RGCs and thus eventually produce a complete accounting of the signals the primate eye sends to the brain.

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

## Methods

### Tissue preparation

Macaque (*M. mulatta*) eyes from animals of either sex (6 males, 5 females, age range 1.17–19.17 years) were obtained immediately post-mortem from the Oregon and California National Primate Research Center biospecimen distribution programs or from UC Berkeley from animals euthanized for unrelated studies. Eyes from UC Berkeley were enucleated under terminal anaesthesia in accordance with procedures approved by the Animal Care and Use Committee of the University of California, Berkeley and as specified in the National Research Council guidelines. The anterior segment was promptly removed before transferring posterior eyecups to Ames' medium equilibrated with 95% oxygen/5% carbon dioxide. Eyecups were stored in the dark at room temperature (20–22 °C) until use (up to 48 h post-mortem). All locations in nasal retina are specified as equivalent eccentricities using the formula $\sqrt{(0.61x)^2 + y^2}$ where $x$ is the distance in mm nasal to the foveal centre and $y$ is the distance in mm superior or inferior to the foveal centre.

Two human retinal samples (male, age 51 and 72 years) were obtained from eyes exenterated for management of orbital tumours at Oregon Health & Science University. No retinal pathology was noted in either case. All samples were de-identified before receipt by the investigators and tissue use was thus deemed non-human subject research by the institutional review board at Oregon Health & Science University.

For calcium imaging experiments in mouse retina, we used either C57BL/6 J (JAX strain: 000664; RRID: IMSR_JAX: 000664) or crossbred Ai95(RCL-GCaMP6f)-D (JAX strain: 028865; RRID: IMSR_JAX:028865) homozygote mice with homozygous Vglut2-ires-Cre knock-in mice (JAX strain: 028863; RRID: IMSR_JAX: 028863), to obtain offspring with GCaMP6f expression in RGCs. Mice were housed under a standard 12:12 h day:night cycle. Ambient temperature and humidity were maintained at 20–22 °C and 50–60%. Mice were of either sex and were 6–26 weeks old. Mice were dark adapted for 1.5 h prior to euthanasia. Animals were anaesthetized with isoflurane and euthanized with cervical dislocation per procedures approved by the Animal Care and Use Committee at University of California, Berkeley. Eyes were then immediately enucleated, the anterior eye removed and the retina isolated for subsequent experiments.

### Immunohistochemistry

The primary and secondary antibodies that were used in this study are detailed in Extended Data Table 1. The specificity of the BNC2 antibody was confirmed using antibodies raised against different epitopes of the same protein and by confirming selective expression of BNC2 in VGLUT3+ amacrine cells in the INL (a pattern predicted by the transcriptome) (Extended Data Fig. 7). Pieces of macaque or human retina were fixed for 30–120 min in 4% paraformaldehyde (PFA) in 0.1 M phosphate buffer at ~22 °C, rinsed in PBS and stored in PBS containing 0.025% NaN₃ (PBS-NaN₃) at 4 °C or cryoprotected in graded sucrose solutions and frozen at −20 °C until use. After washes in PBS, retinas were blocked for 1 h in 1% bovine serum albumin (BSA; Sigma A7030), 150 mM maleimide (Sigma 129585) in PBS. Primary antibodies were diluted in 3% normal horse serum (NHS), 1% Triton X-100, 0.025% NaN₃ in PBS (pH 7.4) and incubated for ~3 days at ~22 °C. Secondary antibodies were diluted in 3% NHS, 0.025% NaN₃ in PBS, and applied overnight at 22 °C. Retinas were counterstained for 20 min in Hoechst 33342 (Invitrogen 33342), then mounted in SlowFade Gold antifade reagent (Invitrogen S36936).

### Fluorescence in situ hybridization

Fluorescence in situ hybridization was performed on horizontal cryosections of the macaque GCL using the RNAscope Multiplex Fluorescent v2 Assay combined with Immunofluorescence–Integrated Co-Detection Workflow (Advanced Cell Diagnostics) according to the manufacturer's instructions. In brief, a macaque retina was fixed with 4% PFA for 24 h at 4 °C, cryoprotected with 10%, 20%, and 30% sucrose and embedded in Cryo-Gel (Leica). A region of peripheral retina was cryo-sectioned through the plane of the GCL at 10 μm thickness. Sections were retrieved for 5 min at -100 °C in Co-detection Target Retrieval solution and then incubated overnight with rabbit anti-RBPMS antibody (Phosphosolutions 1830-RBPMS) to label RGCs. Sections were then digested with Protease III for 30 min at 40 °C and hybridized with probes for macaque *BNC2* (ACD 1238401-C1) and *FSTL4* (ACD 1238411-C2) mRNA. Signals were amplified and fluorescently tagged with 1:1500 TSA Vivid fluorophore 520 (ACD 323271) and 1:1500 TSA Vivid fluorophore 570 (ACD 323272), respectively. Finally, sections were incubated in goat anti-rabbit Alexa Fluor Plus 647 (1:200; Thermo-Fisher A32733) overnight, counterstained with DAPI and mounted in Slowfade Gold.

### Pre-immunolabelling for DiI filling

Pieces of macaque retina were fixed for 30 min in 4% PFA, rinsed in PBS and stored in PBS-NaN₃ at 4 °C until use. For pre-labelling, retinas were incubated on a shaker in primary antibodies for BNC2 and ChAT diluted in incubation buffer containing 3% NHS in PBS-NaN3 for 3 days at 22 °C. After washes in PBS, secondary antibodies were applied in incubation buffer overnight. Some retinas were permeabilized in incubation buffer containing 0.3% Tween 20 for 1 h at 22 °C (3 of 7 cells) before primary antibody incubation. In some cases, secondary detection of ChAT antibodies was performed after dye loading. Pre-labelled retinas were mounted ganglion cell side up on an Olympus BX51WI microscope in PBS-NaN₃. BNC2+ RGCs were located with epifluorescence, then impaled with sharp borosilicate microelectrodes filled with 1% DiI in 100% ethanol under infrared illumination with Dodt or DIC contrast[51,52]. Current pulses (10 s) of ~+10 nA were applied for 3–5 min using a patch-clamp amplifier (HEKA EPC10) for dye injection. Since pre-labelling was with BNC2 antibodies only, we could not unambiguously distinguish pRGC10 and pRGC16 cells. However, the majority of BNC2+ cells are pRGC10 (Fig. 1e) and the most strongly labelled BNC2+ cells are within the pRGC10 cluster (Fig. 1d). On occasion, bistratified cells were recovered that could represent pRGC16 (Extended Data Fig. 8). After cell injections, retinas were post-fixed in PFA overnight at 4 °C, counterstained with Hoescht, mounted in Vectashield (VWR 101098-042) and imaged within one week.

### Fluorescence imaging and image processing

For reconstruction of neuronal morphology, confocal laser scanning and Fast Airyscan images were acquired on a Zeiss LSM 880 microscope with Plan-Apochromat 20×/1.0 DIC water, 20×/0.8 air, and 63×/1.4 oil DIC objectives. Images were acquired at 1.93–2.41 pixels per μm (20×) and 7.6–11.7 pixels per μm (63×) resolution, with optimal axial resolution to permit 3D reconstruction. *Z*-stacks extended from the GCL to the INL, and *z*-step size ranged from 0.65 to 0.86 μm for the 20× objective and 0.16–0.19 μm for the 63× objective. Excitation laser lines were 561 nm for DiI, 488 nm for Alexa-488 (ChAT), 405 nm for Hoescht and 594 nm for Alexa-594 (BNC2). DiI and ChAT were imaged in separate tracks to prevent signal cross-talk.

For confocal imaging of IMHC after calcium imaging or for mosaic analysis, tile scans were acquired with a 20×/1.0 N.A. water or 20×/0.8 N.A. air objective on a Zeiss LSM 880. Tiles of 425.1 × 425.1 μm (2.41 pixels per μm) or 472.33 × 472.33 μm (1.63 pixels per μm) were acquired with 10% overlap and stitched with Zen 2 software (Zeiss) or the Grid/Collections plugins in ImageJ. All modifications to the original images are indicated in the figure legends. A lateral median filter was applied to some images where indicated to remove non-specific signal in the BNC2 channel. In such cases, the filter was applied uniformly to the entire image. Other linear changes to brightness and contrast were applied uniformly to images in ImageJ.

For analysis of the macaque retina hemifield, images were acquired on a Zeiss Axioplan 2 epifluorescence microscope with a Plan Apochromat

10×/0.45 N.A. air objective and a 100 W Hg arc lamp excitation source. Tile scans were acquired using the MosaiX module in Axiovision software and stitched using the Grid/Collection stitching plugin for ImageJ.

## Two-photon calcium imaging

For calcium imaging experiments, pieces of macaque retina (~4–8 mm²) with choroid and retinal pigment epithelium attached were isolated from the sclera under infrared illumination (850 nm). Given the higher density of pRGC10 on the horizontal midline of the retina (Extended Data Fig. 3), most recording fields were from retinal regions extending from the nasal margin of the optic nerve head to the far nasal periphery. Samples were mounted on an inorganic membrane disc (Anodisc, 13 mm, pore size 0.2 μm, GE Whatman) and stabilized with a slice anchor (Warner Instruments). Retinas were oriented in the recording chamber such that the naso-temporal axis was approximately aligned with the horizontal visual stimulation plane. The preparation was continually superfused with warm (36–37 °C) bicarbonate-buffered Ames' medium at 5-6 ml min⁻¹. Samples were imaged with either an Olympus 20×/0.95 N.A. or a Nikon 16×/0.8 N.A. water dipping objective under infrared illumination (850 nm) with oblique contrast optics. Retinas were bolus loaded with the membrane-permeable calcium indicator, Cal-520 AM (dissociation constant ($K_d$) = 320 nM, AAT Bioquest) or Cal-590 AM ($K_d$ = 561 nM, AAT Bioquest). Indicators were dissolved in DMSO containing 20% Pluronic F-127 (ThermoFisher P3000MP), then diluted to a final concentration of 0.91 mM in HEPES-buffered or bicarbonate-buffered Ames' medium (pH = 7.4). For dye loading, a 2–5 MΩ borosilicate pipette was filled with this solution and applied by pressure application (~3 psi for ~10 × 5 s) to the GCL after penetrating the inner limiting membrane. Recordings commenced at least 45 min after dye loading. In some cases, sulforhodamine 101 (0.8 μM; Sigma S7635) was added to the Ames' medium and perfused for 15 min before commencing recordings to visualize the vasculature.

Mouse retinas were dissected off the retinal pigment epithelium under infrared illumination and imaged similarly to macaque. Areas of dorsal hemiretina within 0.7–1 mm of the optic nerve were used to minimize topographical variance in DS tuning[29]. Vglut2-GCaMP6f mice (n = 2) were imaged immediately whereas a wildtype mouse retina was loaded with Cal-590 AM as described above and recording commenced ~1 h after bolus loading.

Two-photon calcium imaging was performed with a modified Scientifica MP-2000 multiphoton microscope and a modified Hyper-Scope multiphoton microscope (Scientifica) fitted with a mode-locked Ti-Sapphire laser tuned to 930 or 1040 nm (Chameleon Ultra II; Coherent). The laser intensity was adjusted so that the minimal power was used to visualize loaded cells (range 9–21 mW for 1,040 nm, 6–25 mW for 930 nm). An open-source CAD model, part list, and filter spectra for both microscope systems can be found at https://github.com/Llamero/Puthussery_Lab_2P_Setup.

**MP-2000 setup.** Two-photon emission light was split by a dichroic mirror (FF552-Di02, Semrock) to red (FF01-590/36, Semrock) and green (FF01-510/42, Semrock) detection channels. The scan field was 256 × 256 pixels or 128 × 128 pixels (220 × 220 μm) acquired at a frame rate of 2.67 Hz or 10.67 Hz. Images were acquired using Sciscan acquisition software (v1.3, Scientifica).

Visual stimuli were generated with a gamma-corrected LG LP097QX1 TFT-LCD retina display (2,048 × 1,536) (Adafruit Industries) that was projected onto the retina through the 20× objective. The RGB output of the display was triple band-pass filtered (FF01-466/555/687-25, Semrock) to separate the RGB visual stimulation bands from the photomultiplier detection channels.

**HyperScope setup.** Two-photon emission light was split by a dichroic mirror (FF552-Di02, Semrock) to red (FF01-590/36, Semrock) and green (FF01-515/30, Semrock) detection channels. The scan field

was 512 × 512 pixels (246.71 × 246.71 μm) acquired at a frame rate of 30 Hz. Images were acquired using ScanImage Basic v2021.01.0 (MBF Biosciences).

Visual stimuli were generated with a fibre-coupled DLP LightCrafter 4500 (EKB Technologies) that was projected onto the retina through the 16x objective. The RGB illumination was generated using XLamp XP-E2 red, cool white, and blue LEDs respectively (CreeLED). The LED light was passed through band-pass filters (FF01-640/20, Semrock, ET550/20×, Chroma, FF01-465/30, Semrock) to separate the RGB visual stimulation bands from the photomultiplier detection channels. LED illumination was controlled via an open-source programmable LED driver (https://github.com/Llamero/Four_Channel_MHz_LED_Driver). Specifically, LED illumination was timed to occur only during the fly-back of the resonant mirror to avoid cross-talk between LED induced excitation of the sample and the two-photon imaging, and the LED channels were synched to the DLP RGB output signal.

**Stimulation protocol.** Light stimuli were generated using custom software written in Igor Pro 9.0 and using PsychoPy toolbox (v2022.2.0 or earlier). Drifting bright bars (500 μm s⁻¹, 200 × 750 μm or 500 × 750 μm; 1.47 × 10⁵ photons s⁻¹ μm⁻² MP-2000, 2.083 × 10⁶ photons s⁻¹ μm⁻² Hyperscope) were presented on a dark background (4.47 × 10³ photons s⁻¹ μm⁻² MP-2000, 3.63 × 10⁴ photons s⁻¹ μm⁻² Hyperscope) and drifted orthogonal to the long axis of the bar at speeds of 500 μm s⁻¹ (corresponding to 2.24 degrees per second using the conversion factor of 223 μm per degree from ref. 53). All angles indicate direction of stimulus motion on the retina. The total stimulation area was 750 μm² (MP-2000) or 1,000 μm² (Hyperscope). Bar direction order was pseudorandomized in blocks such that all directions were presented before repeating and there was an interstimulus interval of 5 s between bar stimuli. A subset of cells were centred in the scan field and probed with bright bars (500 × 750 μm) moving in the preferred and null directions at a range of velocities (125–2,000 μm s⁻¹, 0.57–9 degrees per second). One centred cell was also stimulated with bright spots of increasing diameter (25–750 μm). We allowed 35 s from the laser onset to first stimulus presentation to allow for laser and background adaptation[54]. A translating lens was used to axially offset the visual stimulus from the imaging plane by ~200 μm (for macaque) or ~150 μm (for mouse) so the visual stimulus was focused on the photoreceptors while acquiring two-photon scans of the GCL. For some stimulus protocols, a recording was collected with no laser stimulation for background subtraction of any residual stimulus light reaching the detectors (MP-2000 recordings only).

Imaging procedures for mouse were as for macaque except the drifting bar stimulus was 500 × 750 μm, drifting at 250 μm s⁻¹ (8 degrees per second)[55], a speed that stimulates both ON-type and ON–OFF-type DSGCs in mouse retina[29,34,56,57].

**Pharmacology.** In some experiments, the GABA_A receptor antagonist gabazine (SR95531; HelloBio HB0901) was added to the bath solution (final concentration of 10 μM gabazine for 5 cells, 20 μM for 1 cell). Stock solutions were prepared in $H_2O$ and stored at −20 °C until use. For pharmacological experiments, candidate ON-DSGCs were first identified online by pixel-based vector sum mapping and were centred in the scan field. Calcium responses were then recorded at baseline and at least 4 min after drug wash-in to ensure effects had reached steady-state.

## Data analysis

**Single-cell RNA-sequencing data analysis.** For transcriptomic analysis, we mined existing single-cell RNA-sequencing datasets from macaque (GEO accession GSE118480[14]), human (GEO accession GSE148077[41]), and mouse (GEO accession GSE137400[25]) retina. Cell cluster assignments were as reported in the original publications. Dot plot visualizations were generated using the Broad Institute Single-Cell

Portal (https://singlecell.broadinstitute.org/single_cell) where dot size indicates the proportion of cells in the cluster that expressed the gene and dot colour indicates the relative gene expression level for each row. To identify a candidate ON-DSGC type, we compared expression of GABRA2 and GLRA2 between each peripheral RGC type and its corresponding foveal RGC type. Cell types were excluded as possible candidates if the difference in expression of GABRA2 and/or GLRA2 was >2 log fold and $P < 0.05$ by Wilcoxon rank-sum test.

**Calcium imaging.** Cell somata ROIs of Cal-520 AM- or Cal-590 AM-loaded cells, or GCaMP6f-expressing cells were manually drawn in ImageJ and the average intensity of the ROI over time was extracted for each cell. As detailed above, for some retinas, a no-laser control recording was taken to subtract any low-level stimulus bleedthrough reaching the detectors. For experiments without the no-laser control, a 'background ROI' was drawn in an area containing a blood vessel, with no calcium indicator present, and subtracted from the raw recording.

Raw fluorescence intensity values were imported into Igor Pro (version 9.0.0.10, Wavemetrics) and $\Delta F/F$ values were calculated for each cell using the following equation:

$$\Delta F/F = (F - F_0)/F_0$$

where $F$ is instantaneous fluorescence and $F_0$ is the mean fluorescence over a 23-s window prior to stimulus onset. For some ROIs where there was a slow background oscillation, a high pass filter was applied to restore the baseline. For analysis of responses to drifting bar stimuli, we extracted the time windows when visual stimuli were presented, sorted the directions and averaged the maximum $\Delta F/F$ amplitudes from three trials. The extent of direction selectivity was reported as either the normalized vector sum where the magnitude of the vector sum was divided by the scalar sum of responses to all of the recorded directions (Fig. 2f,g), or in the case of comparison of directional tuning before and after drug application, using the following formula:

$$DSI = \frac{(\Delta F/F_{pref} - \Delta F/F_{null})}{(\Delta F/F_{pref} + \Delta F/F_{null})}$$

The DSI ranges from −1 to 1 with values closer to 1 indicating higher direction selectivity. Negative values indicate a residual response in the opposite direction to the original preferred direction, which was seen in some cells after gabazine application. Only cells that were responsive to the light stimulus, (defined as having an average amplitude >1.5 above baseline s.d.), were included in analysis. Responsive cells were confirmed by manually examining each trace to omit traces where activity was not correlated with the light stimulus. The preferred angles were adjusted so that the superior, temporal, inferior and nasal corresponded to 90, 0, 270, and 180 degrees, respectively. The calcium responses were fit with the von Mises distribution (polar plots in Figs. 2 and 3 and Extended Data Fig. 5), which is the circular analogue of the gaussian distribution, given by:

$$R = \frac{R_{max} \times e^{(\kappa \cos((x-\mu) \cdot \pi/180))}}{e^{\kappa}}$$

where $R$ max is the maximum response, $\mu$ is the preferred orientation in degrees, and $\kappa$ is the width of the tuning curve.

For quantifying calcium responses during bar stimuli moving at different velocities, we measured the maximum $\Delta F/F$ amplitudes over a 0.5-s window during the preferred direction responses. For quantification of responses to different spot sizes, we measured the integral of the $\Delta F/F$ during the 2-s spot stimulation. For the mouse data, group polar graphs and histograms (Extended Data Fig. 5e,f) were generated using custom Python code (Google Colaboratory).

**Pixel-based vector sum mapping.** Potential DSGCs in the scan field were identified by pixel-based vector sum mapping using a custom macro written in ImageJ. In brief, the XYT series was first converted to a $\Delta F/F$ movie and gaussian filtered with a sigma value of 2. Substacks corresponding to each bar presentation period were extracted and z-projected (maximum). The x and y components of the response vectors were then calculated for each pixel in the image and the responses for each angle were averaged. A heat map visualization was used to identify pixels within the scan field showing the highest vector sum values.

**Image registration.** After immunostaining, we acquired an epifluorescence image of retinal pieces used for calcium imaging. The calcium imaging region was identified through matching of vascular landmarks and xy stage reference coordinates. Confocal z-projections of the 2 P scan areas were then acquired and stitched. Confocal images were registered to each 2 P imaging field using custom macros incorporating the bUnwarpJ 2.6.12 plugin in ImageJ (1.53q, NIH). For registration, we made an average z-projection of the 2 P scan field and upsampled the resolution to match the confocal image. To identify $BNC2^+$ and $FOXP2^+$ cells, we set an intensity threshold of 2× the background fluorescence. All scan fields were then manually checked to ensure detection of all immunostained nuclei.

**Tracing DiI-filled cells.** The morphologies of DiI-filled cells were traced from confocal image stacks using the Simple Neurite Tracer (SNT) 4.1.9 plugin in ImageJ. Semi-automated tracing was performed with the A* search algorithm enabled. To fill the skeletonized cells, the distance threshold settings were adjusted manually to best match the confocal image. Somas were traced and filled separately, and their size constrained in cases of oversaturation in the confocal image. In some cases where the connection of the primary dendrites to the soma was not visible due to soma overexposure, the paths were extrapolated using the shortest path. Depth coding of the traced and filled cells was performed by dividing the stack into two layers (GCL to ON-ChAT border and ON-ChAT to INL). Each cell was also inspected to determine the classification of the dendrites in each layer, and manually adjusted to reflect the raw image. Images of cell traces were upsampled (bicubic interpolation) for improved visualization.

**IPL depth and co-fasciculation analysis.** To determine the IPL stratification depth of the DiI-filled cells, z-axis intensity profile plots were generated from regions of the dendritic arbour imaged at 63× (Airyscan). For each cell, 1 or 2 regions were selected, ranging in area from 745 $\mu m^2$ to 17,375 $\mu m^2$. The IPL borders were identified by Hoescht staining, or using the position of the ChAT somas. The average background fluorescence value was subtracted and values were normalized to the maximum fluorescence. Negative values were set to 0. The total IPL depth (average $32.8 \pm 4.3$ $\mu m$) was converted to a percentage value from 0 to 100% (where 0 is the INL border and 100% the GCL border).

The extent of co-fasciculation between the DiI cell fills and ChAT labelling was estimated from 20× images of the complete cell fills. The DiI channel contained the traced and filled binary mask of the cell, and the raw ChAT channel was median filtered and thresholded. The DiI channel was used to mask the ChAT channel to determine the overlapping contact area for each slice in the image stack. The overlapping areas were summed per image and expressed as a percentage of total DiI area. Contact area was measured with the ChAT channel in the normal orientation, and after rotating the ChAT channel about the z-axis by 90 degrees.

**3D surface rendering.** Complete DiI-filled cells were rendered in Imaris 9.8.2 (Oxford Instruments) from cells traced and filled using a manually adjusted threshold in SNT (see 'Tracing DiI-filled cells'). 'Wrap-around' synapses were rendered in Imaris from 63× Airyscan image stacks. The images were first made isotropic in ImageJ by scaling the pixel height and width to match the voxel depth. For both complete DiI cells and

wrap-around synapses, channels were surface rendered in Imaris using default surface settings and manually adjusted thresholds. A cropped region of the DiI dendrite was selected and the ChAT surfaces were filtered to include only those within 1 µm of the DiI dendrite.

**Cell classification, density and mosaic analysis.** We classified BNC2[+] RGCs in tile-scanned confocal $z$-stacks of the macaque GCL taken from four retinas from different animals. Retinal pieces were from peripheral nasal retina between the optic nerve head and far nasal periphery (nasal equivalent eccentricity ~3.5–8.5 mm, average total RGC density 1,251 ± 217 cells per mm$^2$). Scan fields were acquired at an xy resolution of 1.6–2.41 pixels µm$^{-1}$ and tile-scanned with 10% overlap. The stitched image regions that were used for analysis averaged 8.44 ± 5.16 mm$^2$. Image stacks were 2D median filtered to reduce pixel noise and improve visibility of labelled nuclei for subsequent segmentation. The nuclei of BNC2[+]RBPMS[+] cells were manually segmented and checked for accuracy by an additional observer. Intensity measurements were then extracted for the BNC2[+] and FOXP2[+] channels and $z$-scored to permit comparison across samples that had been processed or imaged on different days. FOXP2[+] cells were classified based on a $z$-score threshold intensity value which was cross-checked against manual classification. Cell clusters were independently confirmed using $k$-means clustering, implemented in Igor Pro 9.

For cell density analysis, RGCs were segmented based on RBPMS staining using a semi-automated approach. In brief, a 2D or 3D gaussian blur filter was applied, images were thresholded and objects of interest were detected using automated particle analysis. Alternatively, RGCs were counted by applying a gaussian filter and then using the Find Maxima function in ImageJ. Manual corrections to segmentation were made to correct false positive or false negative segmentation. For spatial cell density plots, images were divided into 500 × 500 µm square sampling regions. In sampling regions where image focus was poor or tissue was damaged or obscured, cell counts were estimated by averaging the surrounding regions. The coverage factor was given by the spatial density of cells (cells per mm$^2$) multiplied by the dendritic field area of that cell type (mm$^2$ per cell). A coverage factor of 1.0 indicates complete tiling with no spaces between arbours or arbour overlap. Higher coverage factors are indicative of greater overlap of the dendritic arbours.

Mosaic analysis was performed on the same regions as described above. Voronoi domain areas were measured for each cell using a custom macro utilizing a built-in ImageJ function (http://biii.eu/voronoi-imagej). The Voronoi domain regularity index (VDRI) was used to assess mosaic regularity and is given by the average Voronoi domain area divided by the standard deviation. Real VDRIs were compared to VDRIs from random simulations of cells at the same density. VDRI values of 1.9 and lower indicate a random array of points[26]. Voronoi domains intersecting with the image boundaries were excluded from analysis. The average VDRI for each retina was calculated before averaging across animals. The density recovery profile (DRP) was determined as described previously[58] and implemented using the sjedrp package in R (sjedrp v0.12, https://github.com/sje30/sjedrp/blob/master/DESCRIPTION; Stephen Eglen). The coordinates of the pRGC10 and pRGC16 centroids were used to calculate the DRP using radii of 0–1,000 µm in 20-µm increments. The DRP profiles from each retinal piece were pooled and mirrored on the $x$ axis to better observe the 'well-like' exclusion zone[27]. DRPs were also calculated for the random simulations. To calculate the approximate convergence, the bottom of the 'well' (average of 3 lowest bins) was taken as a percentage of the average density from 390–1,000 µm (the plateau of the density profile). Data from the first bin were excluded due to elevated variability associated with the smaller area of the innermost annulus[59].

**Fluorescence in situ hybridization analysis.** We identified BNC2[+] FSTL4[+] RGCs in a horizontal section from one macaque retina (superior-nasal region, nasal equivalent eccentricity ~7.4 mm). 16 bit confocal tile scan stacks were acquired with a Plan-Apochromat 20×/0.8 air objective at an xy resolution of 1.93 pixels per µm. For analysis, an area approximately 1.2 mm$^2$ was $z$-projected (sum slices). RGCs were segmented from RBPMS staining using the semi-automated approach described above. The number of puncta per cell was quantified using ACD guidelines (SOP-45-006, ACD). In brief, the average background intensity per fluorophore was determined from a region without cells. This was comparable to the negative control background level. Average intensity per single dot was determined by an average of at least 21 single dots. Then, number of dots were determined per RGC by this formula:

$$\text{Dot number} = \frac{(\text{integrated intensity of } RGC - \text{average background intensity} \times \text{total area of } RGC)}{\text{average intensity per single dot}}$$

After excluding artefacts such as blood vessels or high cell background, cells were considered BNC2[+] if they had more than 40 dots and FSTL4[+] if they had more than 80 dots.

**Statistics and reproducibility.** All data in the figures and text are presented as mean ± s.d. or median and interquartile range unless otherwise indicated. Non-parametric tests were used due to non-normal distribution of data (as determined by the Shapiro–Wilks test) or when sample size was small. For comparisons of independent samples, the Wilcoxon rank-sum test was used. For paired comparisons, the Wilcoxon signed rank test was used. All tests were two-sided and used an alpha level of 0.05 except where multiple comparisons were made. No methods were used to determine sample size a priori. No experimental groups were assigned in this study. Data acquisition and analyses were not performed with blinding to the experimental conditions as most experiments did not involve a treatment or perturbation and analyses were automated. For box plots, interquartile ranges were calculated using the Tukey method. Where representative micrographs are shown, experiments were replicated in multiple retinas as follows: Fig. 1b ($n = 9$ retinas), Fig. 2a ($n = 6$ retinas), Fig. 3a,b ($n = 3$ retinas), Fig. 3e ($n = 3$ retinas), Fig. 4b ($n = 2$ retinas), Extended Data Fig. 1a ($n = 4$ retinas), Extended Data Fig. 5b ($n = 3$ retinas), Extended Data Fig. 6a,b ($n = 6$ retinas, as in Fig. 2), Extended Data Fig. 7a,b,d–f ($n = 2$ retinas) and Extended Data Fig. 8 ($n = 4$ cells from 2 retinas). Micrographs from a single retina are shown in Extended Data Fig. 2d,e.

**Reporting summary**

Further information on research design is available in the Nature Portfolio Reporting Summary linked to this article.

## Data availability

Datasets supporting transcriptomic analyses and figures are available under GEO accessions GSE118480 (macaque), GSE148077 (human), GSE137400 (mouse). Other datasets that were generated in this study are publicly available at Dryad (https://doi.org/10.5061/dryad.47d7wm3kx[60]). Source data are provided with this paper.

## Code availability

Custom code not listed in the methods is available at Dryad (https://doi.org/10.5061/dryad.47d7wm3kx[60]).

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

**Acknowledgements** This work was supported by NIH EY024265 (T.P.), a Hellman Fellows Award (T.P.), a Shaffer Award from the Glaucoma Research Foundation (T.P.), NIH P30 Core Grant P30EY003176 and NIH T32 EY007043. Confocal imaging was conducted at the CRL Molecular Imaging Center, (RRID: SCR_017852), supported by the Helen Wills Neuroscience Institute. The authors thank W. R. Taylor for providing visual stimulus software; W. R. Taylor and M. Feller for feedback on the manuscript; H. Aaron and F. Ives for microscopy advice and support; J. Ng for providing human tissue samples; and J. D. Wallis, the Oregon National Primate Research Center (NPRC) (P51OD011092) and California NPRC (P51OD011107) for providing macaque eyes.

**Author contributions** T.P. and A.Y.M.W. designed the experiments. A.Y.M.W., M.M.K., T.P. and C.F.M. conducted functional experiments. J.G., A.Y.M.W., T.P., A.J.M. and M.H. performed anatomical experiments. A.Y.M.W., T.P., J.G., M.M.K., M.H., Y.Y.Y. and A.J.M. performed data analysis. B.E.S. designed and built instrumentation. T.P. acquired funding. T.P. and A.Y.M.W. wrote the original draft. T.P., A.Y.M.W., M.M.K., B.E.S. and M.H. wrote, reviewed and edited the manuscript.

**Competing interests** The authors declare no competing interests.

**Additional information**
**Correspondence and requests for materials** should be addressed to Teresa Puthussery.

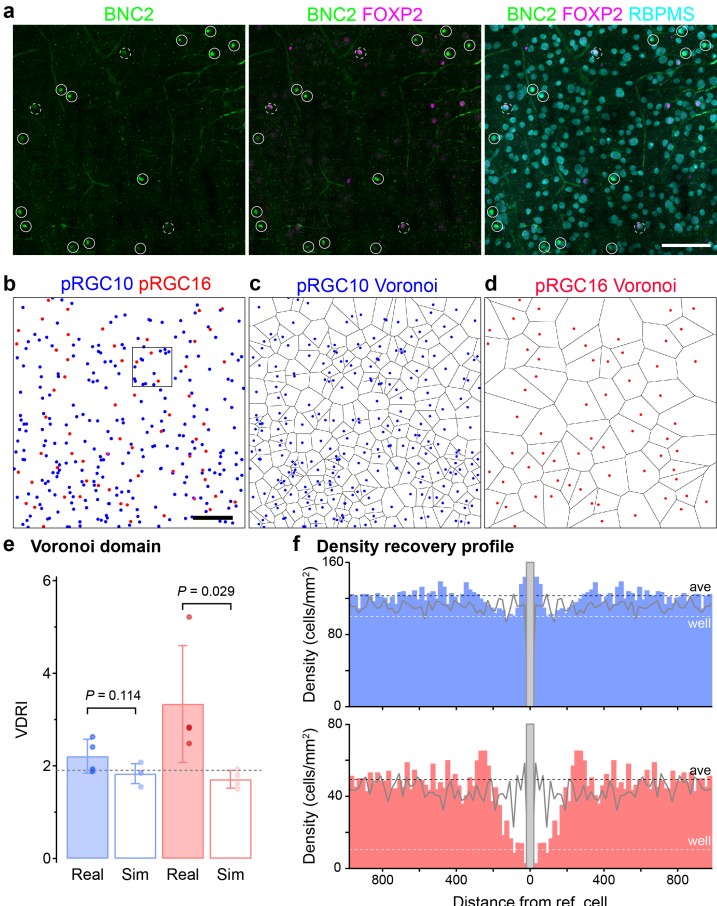

**Extended Data Fig. 1 | Macaque pRGC10 mosaic properties suggest the presence of multiple mosaics. a**, Example confocal image showing BNC2, FOXP2 and RBPMS staining of the macaque GCL with pRGC10 (solid circles) and pRGC16 cells (dashed circles) indicated. Confocal stacks of the GCL were 2D median filtered before max z-projection. **b**, Example of mosaic analysis area showing positions of pRGC10 (blue) and pRGC16 (red) nuclei. Some pRGC10 nuclei are in close proximity suggesting a lack of self-avoidance whereas pRGC16 cells appear more evenly spaced. Dot sizes are not to scale. Region in square ROI is shown in **a. c-d**, Voronoi domain areas for pRGC10 (**c**) and pRGC16 (**d**) from the total area shown in **b. e**, Voronoi domain regularity index (VDRI) for the real and random simulated (sim) mosaics of pRGC10 (blue) and pRGC16 (red) (*n* = 4 retinas). The real pRGC10 mosaics are comparable to random simulations whereas the real pRGC16 mosaic is more regular than its random simulation. The grey dashed line indicates the VDRI of a random array of points (-1.91[26,58],). Data are mean ± s.d. **f**, Density recovery profiles (DRPs) of pRGC10 (blue bars) and pRGC16 (red bars). Densities were pooled from retinal pieces of similar size and RGC density (4 regions from *n* = 3 retinas). Both DRPs show a reduced density at distances closer to the reference cell, seen as a "well-like" zone around each cell in the array where other cells are excluded. The x-axis is mirrored to better visualise this "well". Black dotted lines indicate the average plateau density and white dotted lines mark an average measure of the lowest part of the "well", based on real data. The well will be deepest for a single mosaic and become progressively shallower as more mosaics are added[27,57]. For pRGC16 the density converges at ~21%, which is consistent with the presence of a single mosaic. For pRGC10 the density converges to ~80%, which is consistent with as many as 5 mosaics, however, the data would also be consistent with 3 mosaics, as suggested by the molecular data. Grey lines show DRPs for random simulations with matching numbers of cells and retinal area as the real samples. Overall, the Voronoi domain and DRP analysis indicate that pRGC10 cells have a less regular mosaic structure than pRGC16, consistent with the notion that pRGC10 cells comprise more than one mosaic[27]. Comparisons in **e** were with a two-sided Wilcoxon rank sum test. Scale bars: 100 μm (**a**), 500 μm (**b**).

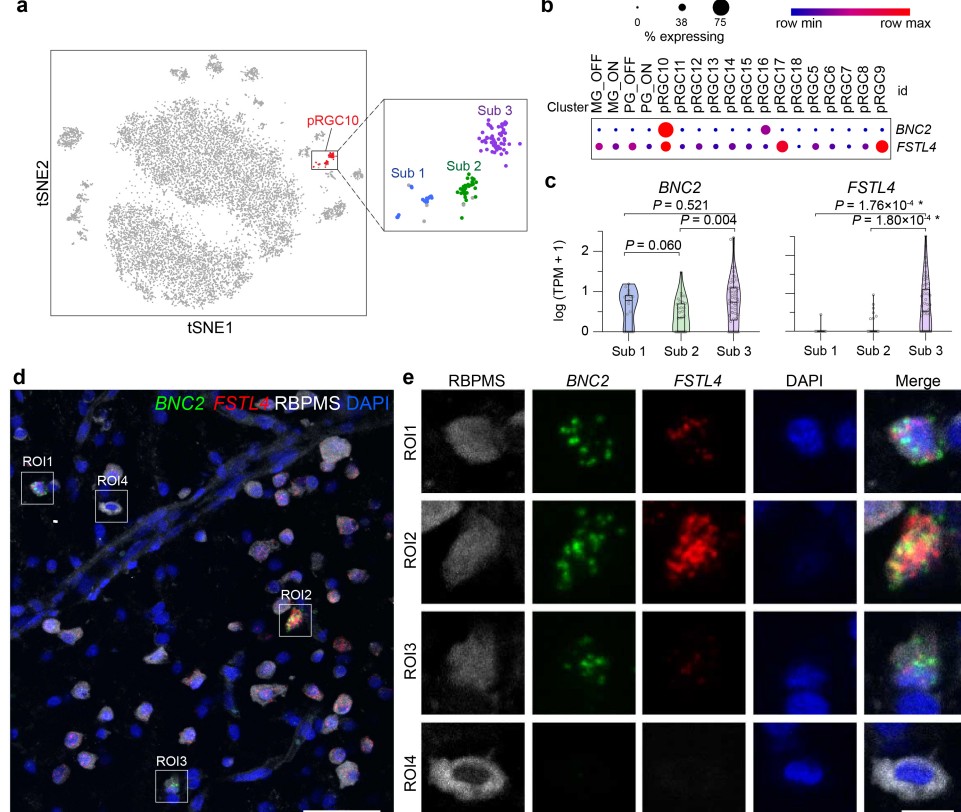

**Extended Data Fig. 2 | pRGC10 cells are comprised of molecular sub-clusters.**
**a**, t-SNE visualisation of macaque peripheral RGC transcriptomic clusters. Raw data were plotted from the dataset of Peng et al., 2019 (GEO: GSE118852)[14]. The pRGC10 cluster (red dots, rectangular ROI) is shown enlarged to the right with putative subclusters (Sub) coloured in blue (Sub 1), green (Sub 2) and purple (Sub 3). Other RGC types are shown by grey dots. **b**, Dot plot showing relative expression of *FSTL4* and *BNC2* in peripheral RGC clusters. **c**, Violin and box/whisker plots comparing expression of *BNC2* and *FSTL4* in the subclusters shown in panel **a**. Boxes cover the interquartile range, line shows median values, whiskers show max and min values. Sub 1 = 55 cells from 4 animals, Sub 2 = 31 cells from 3 animals, Sub 3 = 17 cells from 3 animals. *P*-values for two-sided Wilcoxon rank sum test are shown (corrected for multiple comparisons by Bonferroni method). * Denotes comparisons that were statistically significant by Wilcoxon rank sum and > 2 fold difference in mean expression. Raw data in **a**-**c** from GEO: GSE118852[14]. **d**, Confocal image of a horizontal section of the macaque ganglion cell layer immunolabeled with a pan-RGC marker (RBPMS) and probed by RNA in situ hybridization for *BNC2* and *FSTL4* mRNA. Nuclei are counterstained with DAPI. ROIs 1–4 are enlarged in **e**. **e**, ROI1-ROI3 from panel **d** all express *BNC2*, but only ROI2 shows high levels of *FSTL4* expression. ROI4 is an example of an RGC that lacks expression of both genes. From a total of 594 RGCs, 11 were BNC2+ (1.8%) and 3 of those expressed both *BNC2* and *FSTL4* (27%). Scale bars: 50 μm (**d**), 10 μm (**e**) .

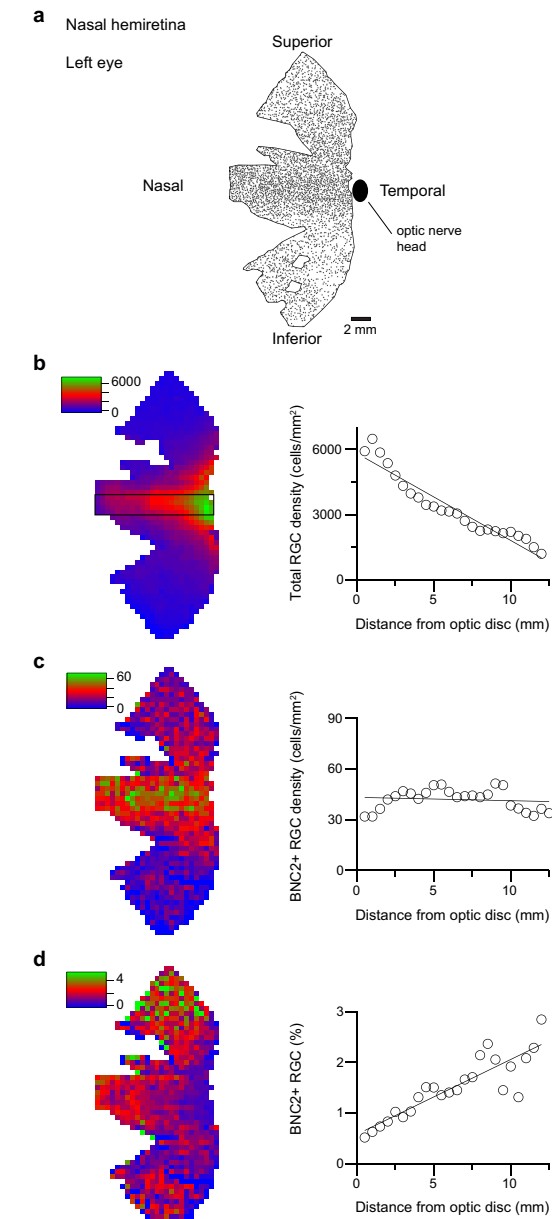

**a**, Nasal hemiretina
Left eye
Superior
Nasal
Temporal
optic nerve head
Inferior
2 mm

**b** 6000 / 0

Total RGC density (cells/mm²)
6000
3000
0
0   5   10
Distance from optic disc (mm)

**c** 60 / 0

BNC2+ RGC density (cells/mm²)
90
60
30
0
0   5   10
Distance from optic disc (mm)

**d** 4 / 0

BNC2+ RGC (%)
3
2
1
0
0   5   10
Distance from optic disc (mm)

**Extended Data Fig. 3 | BNC2 + RGCs are concentrated on the horizontal midline. a**, Nasal half of a macaque retina showing the distribution of BNC2+ RGCs. The black oval shows the approximate position of the optic disc for reference. Scale bar = 2 mm. **b-d**, *Left:* Heatmaps showing total RGC density (**b**), BNC2 + RGC density (**c**) and the percentage of BNC2+RGCs (**d**) for the piece shown in **a**. Note the highest density of BNC2+ cells is on the horizontal midline, whereas the highest percentage of BNC2+ RGCs is in the superior retina. Each box in the heat map represents an area of 500 μm². The black rectangle in **b** delineates the analysis region for the plots shown to the right. *Right*: Plots showing the total RGC density (**b**), BNC2 + RGC density (**c**) and the percentage of BNC2+ RGCs (**d**) as a function of eccentricity for a 2 mm vertical strip centred on the horizontal midline of the retina as indicated in (**b**). Solid lines are linear fits.

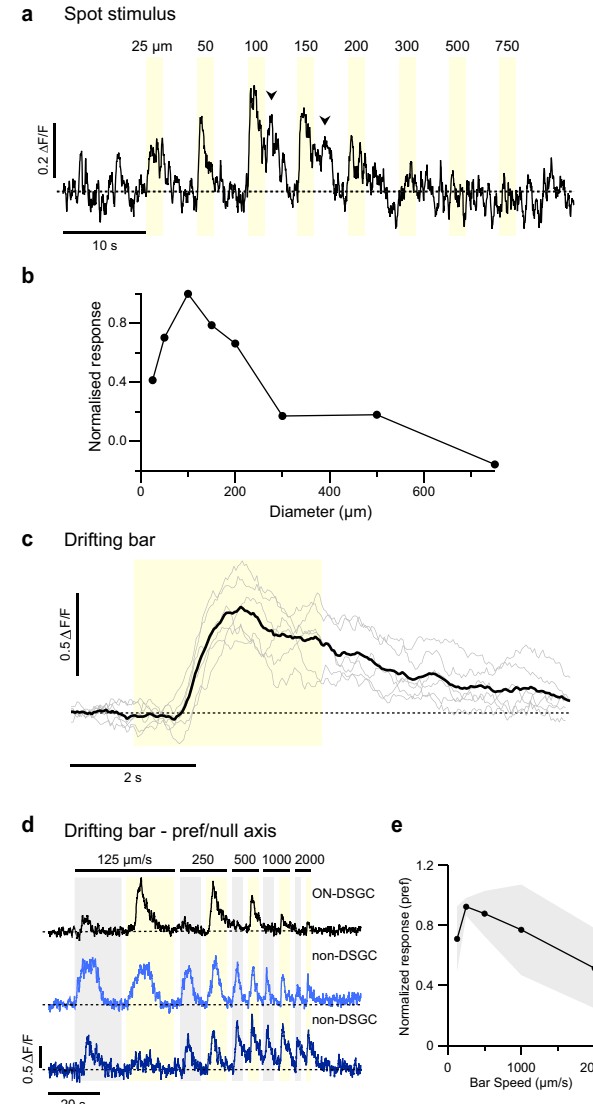

**a** Spot stimulus

**b**

**c** Drifting bar

**d** Drifting bar - pref/null axis

**e**

**Extended Data Fig. 4 | ON-DSGCs show ON responses to spot and bar stimuli and are tuned to slower velocities. a**, Example calcium response of an ON-DSGC (confirmed pRGC10 cell) to bright spots of light presented on a dark background. The cell was positioned in the centre of the scan field and spots were centred on the cell soma. Note the appearance of a small OFF response (arrows) at the termination of the light stimulus for intermediate spot sizes, consistent with reports of sparse OFF bipolar cell input to recursive monostratified RGCs shown previously in macaque[8]. Similar OFF inputs to ON-DSGCs have been reported in other species[29,61]. Spot diameters (μm) are shown above traces. **b**, Normalised ΔF/F integral as a function of spot diameter for the cell shown in **a**. ON-DSGCs generally responded poorly to full-field stimulation presumably due to activation of surround suppression as is shown for this cell. **c**, Response of an ON-DSGC to a bright moving bar stimulus (2.3 °/s) on a dark background. Stimulus timing is shown by yellow shading. The black trace is an average of 6 individual trials (grey traces). Note the sustained ON-phase response but lack of an OFF (trailing edge) response. **d**, Calcium responses of an ON-DSGC to a bar drifting in the null (grey shading) and preferred direction (yellow shading) at velocities ranging from 0.57–9 °/s (125–2000 μm/s). Responses of non-DSGCs from the same scan field to the same stimulus are shown for comparison. **e**, Average normalised peak ΔF/F response as a function of stimulus velocity for four confirmed ON-DSGCs. Responses are from the preferred direction stimulus. Shading shows ±1 s.d.

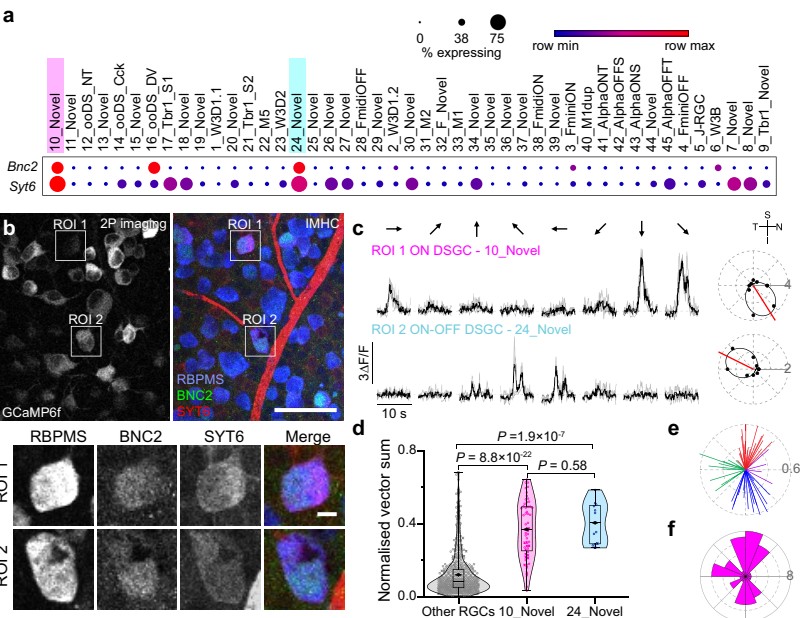

**Extended Data Fig. 5 | Identification of mouse ON-DSGCs with calcium imaging and post hoc immunostaining. a,** Dot plots showing relative expression of *Bnc2* and *Syt6* in mouse RGCs. 10_Novel RGCs (highlighted in magenta) are the predicted ON-DSGC type and 24_Novel (highlighted in cyan) is a predicted ON-OFF DSGC type - both express *Bnc2* and *Syt6*[22,23]. Circle size corresponds to the % of cells expressing the gene and color indicates the relative transcript count in expressing cells. Raw data from GEO: GSE137400[25]. **b,** *Left:* Average z-projection of an example 2-photon calcium imaging scan field showing GCaMP6f in mouse RGCs. *Right*: Same region showing post hoc immunostaining for BNC2, SYT6 and RBPMS. Scale bar = 50 μm. ROI1 and ROI 2 are examples of RGCs that are BNC2 + /SYT6 + /RBPMS + . These ROIs are shown with channels split in *lower* panels and their functional responses are shown in **c.** Scale bar = 5 μm. **c,** *Left:* ΔF/F traces of the ROIs shown in **b** to bars drifting in the 8 directions as shown by arrows above traces (0−360° in 45° increments). ROI1 is an example of a cell classified as an ON-DSGC (molecular cluster Novel_10) and ROI 2 an example of a cell classified as an ON-OFF DSGC

(Novel_24). Grey lines show individual trials, black lines show the average of three trials. Dashed line shows baseline. *Right:* Corresponding polar plots with von Mises fit (smooth black line) and vector sum (red line) for traces shown to the left. The retina is oriented as indicated by the cross (superior (S), nasal (N), inferior (I), temporal (T)). **d,** Box/violin plots showing normalised vector sum of 10_Novel (*n* = 55 cells) and 24_Novel RGCs (*n* = 12 cells) compared to other RGCs (*n* = 473 cells; *n* = 3 mice). Kruskal-Wallis test followed by Dunn's post hoc tests. Boxes cover interquartile range, whiskers indicate minimum and maximum data points, black line indicates median, diamond indicates mean. Open circles indicate data points from GCaMP6f mice, crosses show data points from cells loaded with Cal 590-AM. **e,** Polar plot showing preferred angle (direction of line) and normalised vector sum (length of line) of 10_Novel RGCs. Vectors are coloured according to the cardinal directions with which they most closely align. **f,** Polar histograms summarising the preferred directions of Novel_10 cells, which cluster along superonasal, inferonasal and temporal directions of motion on the retina. Radial axis shows the number of cells.

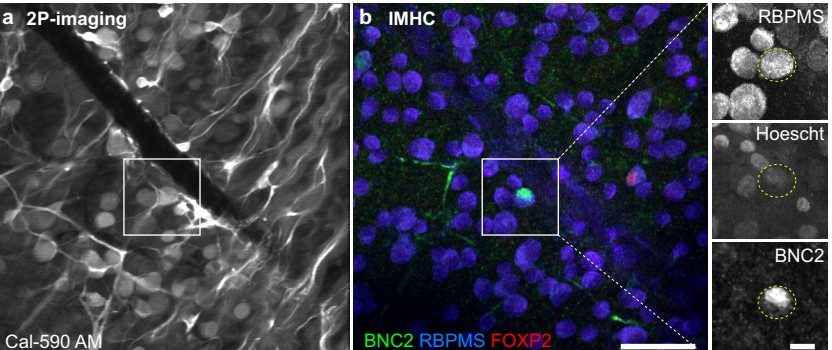

**Extended Data Fig. 6 | Post hoc molecular identification of a macaque ON-DSGC treated with gabazine. a**, Calcium imaging scan field as shown in Fig. 3h. **b**, Post hoc immunostaining of the same area as in **a** for BNC2, FOXP2 and RBPMS confirms the recorded cell is pRGC10 (RBPMS + /BNC2 + /FOXP2−). Rectangular ROI in **b** is shown enlarged with a nuclear stain (Hoescht) on the *right*. Scale bars: main image, 50 µm; inset 10 µm.

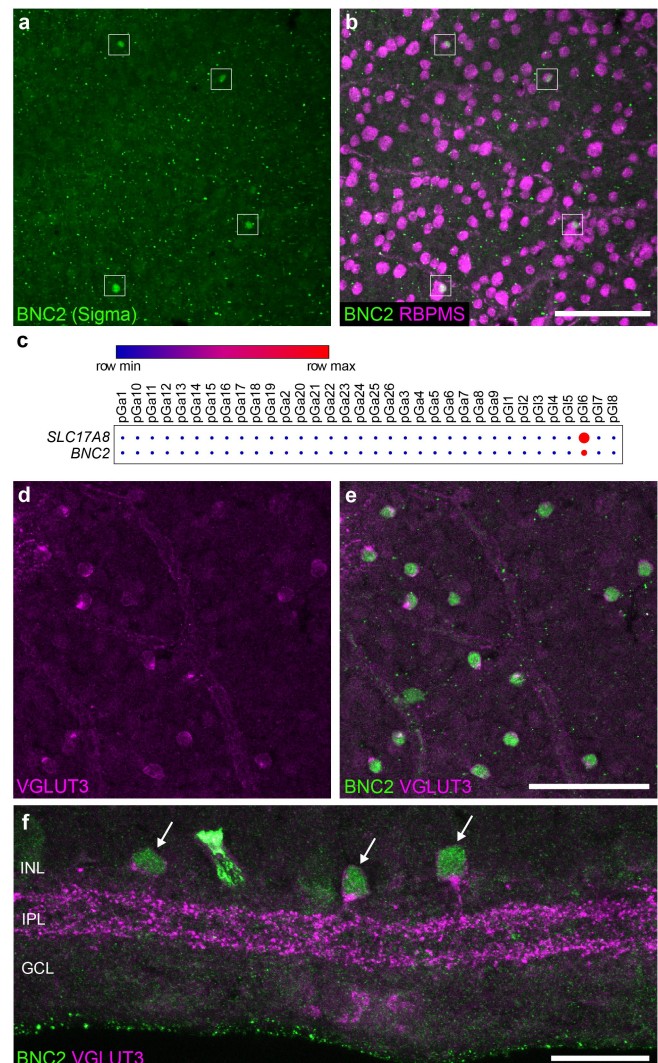

**Extended Data Fig. 7 | Specificity of BNC2 antibodies. a-b**, Macaque flatmount showing GCL labelled for BNC2 (cat.# HPA018525, Sigma) and RBPMS. BNC2 labels the nuclei of a sparse subset of RGCs similar to the pattern seen with rabbit anti-BNC2 (cat.# PA584417, Invitrogen). Square ROIs show examples of BNC2+ cells. **c**, Dot-plot showing relative expression of *SLC17A8* (VGLUT3) and *BNC2* in peripheral glycinergic amacrine (pGl) cells. High expression of both genes is evident in pGl6. Raw data from GEO: GSE118852[14]. **d-e**, Wholemount of macaque retina showing immunolabelling for BNC2 and VGLUT3 at the level of the inner nuclear layer. **f**, Vertical section of macaque retina showing amacrine cells that express both BNC2 and VGLUT3. The same cells are labelled with both antibodies in **e** and **f**, consistent with the transcriptomic data in **c**. Arrows in **f** show examples of double labelled cells. Scale bars: 100 μm (**a-b**); 50 μm (**d-f**).

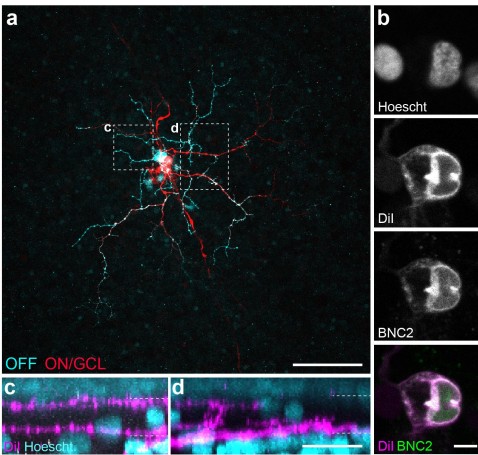

**Extended Data Fig. 8 | BNC2+ cells with bistratified morphology.**
**a**, Z-projection (maximum) of a bistratified BNC2+ cell filled with DiI. The fill is
depth coded to highlight OFF (cyan) and ON (red) dendrites in the IPL. **b**, Same
cell as in **a** showing BNC2+ staining in the nucleus (demarcated by Hoescht
staining). Note that some crosstalk of the DiI signal is present in the BNC2
channel, but the nuclear pattern of BNC2 staining is evident with the DiI and
BNC2 channels merged. **c-d**, Side projections of the boxed areas in **a** showing
ON and OFF dendrites in the IPL. The borders of the IPL are demarcated by dotted
lines based on Hoescht staining. Scale bars:100 µm (**a**), 5 µm (**b**), 20 µm (**c**, **d**).

## Extended Data Table 1 | Details of antibodies used in this study

| Antibody | Host | Source | Cat# / Lot # | Dilution | Immunizing antigen | Validation |
|---|---|---|---|---|---|---|
| RBPMS<br>RRID:AB_2492226 | Guinea pig | PhosphoSolutions | Cat: 1832-RBPMS<br>Lot: NB919g | 1:500 | Synthetic peptide corresponding to amino acid residues from the N-terminal region of rat RBPMS, conjugated to keyhole limpet hemocyanin (KLH). | See antibody registry: AB_2492226 |
| RBPMS<br>RRID:AB_2492225 | Rabbit, | PhosphoSolutions | 1830-RBPMS<br>Lot: NB521y | 1:100 | Synthetic peptide corresponding to amino acid residues from the N-terminal region of rat RBPMS, conjugated to keyhole limpet hemocyanin (KLH). | See antibody registry: AB_2492226 |
| BNC2<br>RRID:AB_2791569 | Rabbit | Invitrogen; Thermo Fisher Scientific | Cat: PA5-84417<br>Lots: WC3226213A, WH3349141B, WK3443178B | 1:50 | Immunogen sequence: STQNEYNESS ESEVSPTPYK NDQTPNRRNAL TSITNVEPKT EPACVSPIQN SAPVSDLTKT EHPKSSFRIH RMRRMGSASR KGRVFCNA | Expected pattern based on transcript expression – this study. Same pattern as Sigma HPA018525. |
| BNC2<br>RRID:AB_1845402 | Rabbit | Atlas Antibodies/ Sigma-Aldrich | Cat: HPA018525<br>Lot: A45570 | 1:500-1:1500 | Zinc finger protein basonuclin-2 recombinant protein epitope signature tag (PrEST) | This study; See also Human Protein Atlas |
| ChAT<br>RRID:AB_2079751 | Goat | Millipore | Cat: AB144P<br>Lot: 2620146 | 1:50 | Human placental enzyme. | See antibody registry: AB_2079751 |
| FOXP2<br>RRID:AB_1268914 | Goat | Abcam | Cat: Ab1307<br>Lot: GR3270597-12 | 1:1000 | Synthetic peptide corresponding to Human FOXP2 aa 703-715 (C terminal) (Cysteine residue). Sequence: C-REIEEEPLSEDLE | Rousso et al., 2016; Tran et al., 2019 |
| Synaptotagmin-6 (SYT-6)<br>RRID: AB 11001830 | Mouse | Antibodies Incorporated | Cat: 75-271<br>Lot:449-1AK-44 | 1:1000 | Fusion protein amino acids 246-333 (cytoplasmic C2A domain) of mouse Synaptotagmin-6 | See antibody registry: AB_11000182 |
| VGLUT3<br>RRID:AB_2187832 | Guinea pig | Millipore | Cat: AB5421<br>Lot: 2728274 | 1:4000 | KLH-conjugated linear peptide corresponding to a C-terminal cytoplasmic domain sequence of rat vesicular glutamate transporter 3 (VGluT3) | Expected pattern based on transcript expression (this study). Also see RRID. |
| Rabbit Alexa Fluor 488<br>RRID:AB_2535792 | Donkey | Molecular Probes | Cat: A-21206<br>Lot: 1910751 | 1:800 | Gamma Immunoglobins (Heavy and Light chains) | |
| Guinea Pig Alexa Fluor 594<br>RRID:AB_2340474 | Donkey | Jackson ImmunoResearch | Cat: 706-585-148 | 1:500 | Gamma Immunoglobins (Heavy and Light chains) | |
| Goat Alexa Fluor 647<br>RRID:AB_141844 | Donkey | Molecular Probes | Cat: A-21447<br>Lot:737682 | 1:800 | Gamma Immunoglobins (Heavy and Light chains | |
| Rabbit Alexa Fluor Plus 647<br>RRID: AB_2633282 | Goat | Invitrogen | Cat: A-32733 | 1:200 | Gamma Immunoglobins (Heavy and Light chains) | |

# Reporting Summary

## Statistics

For all statistical analyses, confirm that the following items are present in the figure legend, table legend, main text, or Methods section.

| n/a | Confirmed | |
|---|---|---|
| ☐ | ☒ | The exact sample size (*n*) for each experimental group/condition, given as a discrete number and unit of measurement |
| ☐ | ☒ | A statement on whether measurements were taken from distinct samples or whether the same sample was measured repeatedly |
| ☐ | ☒ | The statistical test(s) used AND whether they are one- or two-sided<br>*Only common tests should be described solely by name; describe more complex techniques in the Methods section.* |
| ☒ | ☐ | A description of all covariates tested |
| ☐ | ☒ | A description of any assumptions or corrections, such as tests of normality and adjustment for multiple comparisons |
| ☐ | ☒ | A full description of the statistical parameters including central tendency (e.g. means) or other basic estimates (e.g. regression coefficient) AND variation (e.g. standard deviation) or associated estimates of uncertainty (e.g. confidence intervals) |
| ☐ | ☒ | For null hypothesis testing, the test statistic (e.g. *F*, *t*, *r*) with confidence intervals, effect sizes, degrees of freedom and *P* value noted<br>*Give P values as exact values whenever suitable.* |
| ☒ | ☐ | For Bayesian analysis, information on the choice of priors and Markov chain Monte Carlo settings |
| ☒ | ☐ | For hierarchical and complex designs, identification of the appropriate level for tests and full reporting of outcomes |
| ☒ | ☐ | Estimates of effect sizes (e.g. Cohen's *d*, Pearson's *r*), indicating how they were calculated |

*Our web collection on statistics for biologists contains articles on many of the points above.*

## Software and code

Policy information about availability of computer code

| Data collection | Zen 2 software (Zeiss), SciScan (v1.3, Scientifica), Zeiss Axiovision software (v 4.8.0.0), ScanImage Basic v2021.01.0 (MBF Biosciences), PsychoPy 2022.2.0 or earlier |
|---|---|
| Data analysis | ImageJ (1.53q, NIH, USA), Igor Pro (version 9.0.0.10, Wavemetrics), Imaris 9.8.2 (Oxford Instruments), sjedrp v0.12, Simple Neurite Tracer Plugin for ImageJ (4.1.9), bUnwarpJ 2.6.12 plugin for ImageJ, custom ImageJ macros, custom Python code (Google Colaboratory) |

For manuscripts utilizing custom algorithms or software that are central to the research but not yet described in published literature, software must be made available to editors and reviewers. We strongly encourage code deposition in a community repository (e.g. GitHub). See the Nature Portfolio guidelines for submitting code & software for further information.

## Data

Policy information about availability of data

All manuscripts must include a data availability statement. This statement should provide the following information, where applicable:
- Accession codes, unique identifiers, or web links for publicly available datasets
- A description of any restrictions on data availability
- For clinical datasets or third party data, please ensure that the statement adheres to our policy

GEO datasets: GEO:GSE118480 (macaque), GEO:GSE148077 (human), GEO:GSE137400 (mouse)
All raw data generated in this study has been deposited in a publicly available data repository.

# Human research participants

Policy information about studies involving human research participants and Sex and Gender in Research.

| | |
|---|---|
| Reporting on sex and gender | N/A |
| Population characteristics | N/A |
| Recruitment | N/A |
| Ethics oversight | N/A |

Note that full information on the approval of the study protocol must also be provided in the manuscript.

# Field-specific reporting

Please select the one below that is the best fit for your research. If you are not sure, read the appropriate sections before making your selection.

☒ Life sciences          ☐ Behavioural & social sciences          ☐ Ecological, evolutionary & environmental sciences

For a reference copy of the document with all sections, see nature.com/documents/nr-reporting-summary-flat.pdf

# Life sciences study design

All studies must disclose on these points even when the disclosure is negative.

| | |
|---|---|
| Sample size | No statistical methods were used to determine sample size a priori. Sample size was determined based on similar experiments on sparse primate retinal ganglion cell types. We used appropriate statistical tests to determine statistical significance given the sample size. |
| Data exclusions | For calcium imaging, cells that were deemed unresponsive based on a response threshold (dF/F < 1.5 s.d. above baseline) were excluded from further analysis. |
| Replication | All experiments were repeated on multiple cells from different animals as indicated in the manuscript. |
| Randomization | No experimental groups were assigned in this study. |
| Blinding | Data acquisition and analyses were not performed with blinding to the experimental conditions as most experiments did not involve a treatment or perturbation and analyses were automated. In the case of experiments using gabazine, the control and drug trials were analyzed automatically, without consideration of trial conditions. |

# Reporting for specific materials, systems and methods

We require information from authors about some types of materials, experimental systems and methods used in many studies. Here, indicate whether each material, system or method listed is relevant to your study. If you are not sure if a list item applies to your research, read the appropriate section before selecting a response.

## Materials & experimental systems

| n/a | Involved in the study |
|---|---|
| ☐ | ☒ Antibodies |
| ☒ | ☐ Eukaryotic cell lines |
| ☒ | ☐ Palaeontology and archaeology |
| ☐ | ☒ Animals and other organisms |
| ☒ | ☐ Clinical data |
| ☒ | ☐ Dual use research of concern |

## Methods

| n/a | Involved in the study |
|---|---|
| ☒ | ☐ ChIP-seq |
| ☒ | ☐ Flow cytometry |
| ☒ | ☐ MRI-based neuroimaging |

## Antibodies

| | |
|---|---|
| Antibodies used | Full details of antibodies used in this study have been provided with the submission in Extended Data Table 1 |
| Validation | Full details of antibody validation has been provided in Extended Data Table 1. Data supporting validation of BNC2 antibodies are |

| Validation | provided in Extended Data Fig. 7. We tested multiple antibodies to compare localization patterns and confirmed that expression matched that expected based on orthogonal methods (single-cell RNA sequencing). |
|---|---|

# Animals and other research organisms

Policy information about <u>studies involving animals</u>; <u>ARRIVE guidelines</u> recommended for reporting animal research, and <u>Sex and Gender in Research</u>

| Laboratory animals | Mus muscularis: B6J.Cg-Gt(ROSA)26Sortm95.1(CAG-GCaMP6f)Hze/MwarJ, (JAX strain #:028865; RRID:IMSR_JAX:028865), B6J.129S6(FVB)-Slc17a6tm2(cre)Lowl/MwarJ, (JAX strain #028863; RRID:IMSR_JAX:028863), C57BL/6J (JAX strain#:000664; RRID:IMSR_JAX:000664). Age: 6 - 26 weeks. Animals had ad libitum access to food and water and were kept on a 12/12 h light/dark cycle. Ambient temperature and humidity were maintained at 20-22°C and 50-60%.<br>Rhesus Macaque: Macaca mulatta. Age 1.36-19.17 years. |
|---|---|
| Wild animals | No wild animals were used in this study. |
| Reporting on sex | Macaque eyes were obtained from 6 males and 5 females. Mouse eyes were from 1 male and 2 females. Data were not disaggregated for sex given the small number of animals used in each experiment. |
| Field-collected samples | There were no field-collected samples in this study. |
| Ethics oversight | All mouse procedures were approved by the Animal Care and Use Committee of the University of California, Berkeley. Macaque tissue collection procedures at UC Berkeley were approved by the UC Berkeley Animal Care and Use Committee and conducted in accordance with National Research Council guidelines. Tissues from Oregon & California National Primate Centers were collected immediately post-mortem from animals used for unrelated studies. |

Note that full information on the approval of the study protocol must also be provided in the manuscript.

