## [Peer Review File · Nature]

Manuscript Title: An ON-type direction selective ganglion cell in primate retina

Reviewer Comments & Author Rebuttals

Reviewer Reports on the Initial Version:

Referees' comments:

Referee #1 (Remarks to the Author):

Summary

In this manuscript, Wang et al. provide the first characterization of ON direction selective ganglion cells (ON DSGCs) in primate retina. To do so, they identify molecular markers of putative ON DSGCs and show that the morphological and mosaic characteristics of these cells are similar to what has been observed in mouse ON DSGCs. Next, they use calcium imaging to show that their putative ON DSGCs are direction selective, and that this tuning may rely on GABAergic transmission, as is the case in other species. Finally, they identify a possible human homologue of ON DSGCs. This work is important, not only because it reveals the existence of a new cell type in primates, but also because it demonstrates an approach for identifying primate and human homologous of circuits that have been characterized in other species. However, the claim that the authors have identified the primate homologue of ON DSGCs requires further substantiation to be fully convincing. Further, the interpretability of the data would benefit greatly from increasing the sample size in several experiments.

Major Points

- The central argument of this manuscript is that pRGC10s are the primate homologue of ON DSGCs involved in the accessory optic system. While the authors have generated evidence that is consistent with this claim, the data are not entirely convincing. Further experiments would help assuage these concerns:

- o Back labeling: To make the connection with the accessory optic system convincing would require (1) retrogradely labeled ganglion cells from the accessory optic system nuclei, as was done previously (Patterson et al., 2022); (2) physiological evidence of direction selectivity; and (3) immunostaining for the molecular markers identified in this manuscript. Currently, the manuscript has no evidence for (1), weak evidence for (2) (see comment below), and strong evidence for (3). The connection to the accessory optic system is critical to the potential significance of this work.

- o Step Response: If pRGC10s are ON DSGCs, they should have an ON response to a step of light. The authors show that pRGC10s primarily stratify in the ON sublamina of the IPL and have a unimodal response to a drifting bar of light. However, a more direct and convincing way to test the polarity of pRGC10s is to measure their response to a full-field step of light. This is a critical experiment for differentiating pRGC10s from ON-OFF and OFF DSGCs that have been identified in other mammals.

- o Direction Selectivity: It is difficult to assess whether pRGC10s are indeed direction selective. Figure 2f shows that while they are significantly more direction selective than other RGCs, their DSIs are still quite low compared to previous descriptions of both ON and ON-OFF DSGCs (e.g. Sabbah et al., 2017 who also used calcium imaging). Further, the preferred directions of ON DSGCs are known to cluster into 3 (or maybe 4) directions (Oyster and Barlow, 1967; Sabbah et al., 2017), but Figure 2g shows that there is wide variability in the preferred directions of pRGC10s. The analogous data from mice in Extended Data Figure 4d, e present similar concerns.

To overcome these challenges, the authors could first establish a benchmark by demonstrating that they can reliably detect direction selective responses from known populations of DSGCs in mouse. This can be achieved by performing the same calcium imaging experiments on transgenic lines in which ON (*Hoxd10*, *Spig1*) or ON-OFF (*Drd4*, *Trhr*) DSGCs are labeled, or by retrogradely labeling ON DSGCs via central injection into accessory optic nuclei. This experiment would clarify the magnitude of DSI and variance of preferred direction to expect given the authors' experimental paradigm. It would also facilitate comparison to previous work on DSGCs. Second, the authors should consider increasing the number of replicates performed on the pRGC10s in order to solidify the validity of their results given the large variance and small effect size in Figure 2f.

o Speed Tuning: ON DSGCs in lower mammals have both direction and speed tuning (Oyster et al., 1972.; Sivyer et al., 2019; Summers and Feller 2022). Together, these characteristics are critical to their role in subserving the optokinetic reflex and differentiate them from other RGC types. The authors should therefore test the speed tuning of pRGC10s. If pRGC10s are tuned to slow stimuli, it would greatly strengthen the claim that they are homologous to ON DSGCs. This would also validate the authors' choice to use glycine receptor subunits to identify pRGC10s in the first place.

- Mosaic Analyses: The authors claim that pRGC10 somas were randomly distributed across the retina, indicating that this cluster consists of multiple RGC types. However, multiple overlapping mosaics should not make a random spatial distribution. The authors should clarify this point. Further, it would be valuable to know how many independent mosaics exist within the pRGC10 cluster and to relate this to the number of ON DSGCs that have been identified in other species. The number of constituent mosaics in the pRGC10 cluster can be computed by examining finer details of the spatial autocorrelations and density recovery profiles (Cook and Podugolnikova, 2001). This is especially important given the wide variance in preferred direction shown in figure 2f. It would also be valuable to perform mosaic analyses for the BNC2+ human RGCs in order to provide more convincing evidence that these cells are the same as those identified in the macaque.

- GABAzine Experiments: The authors assert that their data "indicate that directional selectivity in primate ON-DSGCs depends critically on GABAergic inhibition from SACs." This claim is unsubstantiated by the data. First, the GABAzine experiment in figure 3 requires further replicates (current sample size is 2 cells) in order to be properly interpreted. More concerning, however, is that the current data do not suggest the direct involvement of SACs. One would expect that if direction selectivity in pRGC10s relied on SAC input, GABAzine would increase responses to null direction stimuli and leave the responses to preferred direction stimuli relatively unchanged. The data in figure 3g do not confirm this prediction: In the first cell, GABAzine appears to decrease the response to preferred direction stimuli while leaving null direction stimuli relatively unchanged. In the second cell, GABAzine does not appear to abolish direction selectivity, but simply changes the cell's preferred direction. The experiment does not support the authors' intentions to show that the circuitry underlying the direction selectivity is similar to the SAC circuit shown in other species. Otherwise, additional experiments would be required to clarify the contribution of SACs to pRGC10 direction selectivity, such as increasing the sample size and perhaps patching directly from pRGC10s to measure the extent to which inhibitory inputs are direction selective.

Minor Points

- In some species, AOS-projecting RGCs are displaced, with their soma in the inner nuclear layer. The authors should comment on whether any fraction of pRGC10s are displaced.

- Line 31: "In lower mammals, ON DSGCs are the first site of directional computation...". The first site of direction selectivity is the SAC. This point should be clarified, especially because figure 3 examines direction selective inhibition, presumably from SACs, to ON DSGCs.

- Von Mises fits in figure 2e and extended figure data figure 4c are not very helpful and it is not

clear what they add considering the tuning curves of these cells tend to not be very symmetric.

- The authors should report the speed of the drifting bars used on primate tissue in degrees per second and define its contrast and absolute light intensity.
- Line 150: "These data strongly suggest that a functionally homologous ON-DSGC is present in the human retina." This statement is overly strong and would require substantially more experimentation to justify. Consider removing the word "strongly".

References

Cook, J. E., & Podugolnikova, T. A. (2001). Evidence for spatial regularity among retinal ganglion cells that project to the accessory optic system in a frog, a reptile, a bird, and a mammal. *Visual neuroscience*, 18(2), 289-297.

Oyster, C. W., & Barlow, H. B. (1967). Direction-selective units in rabbit retina: distribution of preferred directions. *Science*, 155(3764), 841-842.

Oyster, C. W., Takahashi, E., & Collewijn, H. (1972). Direction-selective retinal ganglion cells and control of optokinetic nystagmus in the rabbit. *Vision research*, 12(2), 183-193.

Patterson SS, Bembry BN, Mazzaferri MA, Neitz M, Rieke F, Soetedjo R, Neitz J. Conserved circuits for direction selectivity in the primate retina. *Curr Biol*. 2022 Jun 6;32(11):2529-2538.e4.

Sabbah, S., Gemmer, J. A., Bhatia-Lin, A., Manoff, G., Castro, G., Siegel, J. K., ... & Berson, D. M. (2017). A retinal code for motion along the gravitational and body axes. *Nature*, 546(7659), 492-497.

Sivyer, B., Tomlinson, A., & Taylor, W. R. (2019). Simulated saccadic stimuli suppress ON-type direction-selective retinal ganglion cells via glycinergic inhibition. *Journal of Neuroscience*, 39(22), 4312-4322.

Summers, M. T., & Feller, M. B. (2022). Distinct inhibitory pathways control velocity and directional tuning in the mouse retina. *Current Biology*.

Referee #2 (Remarks to the Author):

The manuscript by Wang and Colleagues identifies, for the first time, ON-direction selective retinal ganglion cells (ON DSGCs) in the non-human primate retina and provides compelling evidence for their presence in the human retina.

Background and significance: The retinas of mice, rabbits, and other prominent vertebrate models exhibit many ganglion cells that are direction selective. However, the presence of DSGCs in primates has remained elusive. It has been broadly assumed that DSGCs are absent from the primate retina, and cortical circuits that compute directional signals de novo are the exclusive source of explicit representation of motion direction in the primate brain. At tension with this view were two observations. First, there remain many morphologically defined ganglion cell types whose function is unexamined. Second, the primate retina contains starburst amacrine cells, a critical component in the production of DS signals in the retinas of all 'lower' vertebrates. The conclusive discovery of ON DSGCs in the primate retina, and their genetic, morphological, and physiological similarity to the ON DSGCs in lower vertebrates, is highly significant for our understanding of primate (including human) visual processing and for our understanding of the

evolution of vision across vertebrates. In my estimation this is one of the most significant discoveries related to the retina and early visual processing over the last decade.

Data and Methodology: The approach developed here of leveraging transcriptional signatures of ON DSGCs from the mouse to identify candidate ON DSGCs from primate transcriptomics data is ingenious. Overall, the methodological approaches used in this manuscript are all executed to the highest standards and are clearly presented.

Robustness and Reliability: The confluence of gene expression, morphology, circuit connectivity, physiology and pharmacology, provide an exceedingly convincing case that support the discovery of ON DSGCs in the primate retina.

Clarity and Scholarliness. The manuscript is very clearly written. The figures present exactly what a reader wants to know. The prior literature is well cited.

Suggested improvements. This is probably the most complete and solid manuscript for an initial submission that I have had the pleasure of reviewing. It reads like a manuscript that has already undergone a round or two of reviews. Many controls are already present, and I struggle to identify additional experiments or analyses that I would want to see. If I were to nit-pick, it would be very nice if the author could show, or at least comment on whether they think there are 3 or 4 directions (mosaics) of these ON DSGCs. As the authors are no doubt familiar, recent work from the Berson lab indicates the presence of 4 ON DSGC types with 4 different direction preferences, while prior work from Oyster and colleagues in the rabbit indicates just 3. One perspective is that the axes of the ON DGCS are registered to the axes of self-motion (Berson and colleagues), while another perspective is that they are registered to the axis of rotation represented by the semi-circular canals (Simpson and Soodak). But this is only a suggestion for a slight improvement; the study and manuscript are already quite impressive.

Author Rebuttals to Initial Comments:

We thank the reviewers for the constructive and helpful suggestions to improve the manuscript. We outline our responses to each of the concerns in blue below.

Referees' comments:

Referee #1 (Remarks to the Author):

Summary

In this manuscript, Wang et al. provide the first characterization of ON direction selective ganglion cells (ON DSGCs) in primate retina. To do so, they identify molecular markers of putative ON DSGCs and show that the morphological and mosaic characteristics of these cells are similar to what has been observed in mouse ON DSGCs. Next, they use calcium imaging to show that their putative ON DSGCs are direction selective, and that this tuning may rely on GABAergic transmission, as is the case in other species. Finally, they identify a possible human homologue of ON DSGCs. This work is important, not only because it reveals the existence of a new cell type in primates, but also because it demonstrates an approach for identifying primate and human homologous of circuits that have been characterized in other species. However, the claim that the authors have identified the primate homologue of ON DSGCs requires further substantiation to be fully convincing. Further, the interpretability of the data would benefit greatly from increasing the sample size in several experiments.

We thank the reviewer for acknowledging the importance of this work. We have now performed new experiments and analyses that we feel further substantiates our conclusions as detailed below. We have also increased sample size for several experiments.

Major Points

- The central argument of this manuscript is that pRGC10s are the primate homologue of ON DSGCs involved in the accessory optic system. While the authors have generated evidence that is consistent with this claim, the data are not entirely convincing. Further experiments would help assuage these concerns:

- o Back labeling: To make the connection with the accessory optic system convincing would require (1) retrogradely labeled ganglion cells from the accessory optic system nuclei, as was done previously (Patterson et al., 2022); (2) physiological evidence of direction selectivity; and (3) immunostaining for the molecular markers identified in this manuscript. Currently, the manuscript has no evidence for (1), weak evidence for (2) (see comment below), and strong evidence for (3). The connection to the accessory optic system is critical to the potential significance of this work.

The morphology of the ON-DSGCs in our study is strikingly similar to the recursive monostratified cells filled previously by retrograde labeling from the NOT-DTN (see Fig. 6D of (Patterson et al., 2022) and from the pretectum (reported to include NOT-DTN) by Kim et al., 2022. Moreover, prior studies have shown retrogradely labeled RGCs from the macaque NOT-DTN (Telkes et al., 2000), anterograde labeling from retina to MTN (Itaya and Van Hoesen, 1983) and the presence of directional responses in NOT-DTN cells prior to development of cortical inputs to this brain region (a result interpreted to mean that directional inputs were coming directly from the retina; Distler and Hoffmann, 2001). The most parsimonious explanation is that the cells identified in our study project to the AOS, like ON-DSGCs in other vertebrates.

Although the proposed experiments combining retrograde labeling, calcium imaging and immunostaining might provide additional proof of the connection to the AOS, we do not have access to the monkeys, equipment or expertise to conduct these experiments. Moreover, the technical challenges and complexities associated with such experiments would mean a high likelihood of failure. For example, Patterson et al., 2022 apparently retrogradely labeled these cells, but were unable to successfully record from them, despite their considerable expertise with both cortical injections and isolated primate retina recording. Moreover, cleanly targeting the NOT-DTN for tracer injection without hitting fibers *en route* to the superior colliculus is non-trivial, as is preventing tracer spread to neighboring pretectal nuclei - both of which could confound interpretation. We hope that our efforts described below to address the reviewer's other comments will help assuage concerns about the interpretation of our findings.

- o Step Response: If pRGC10s are ON DSGCs, they should have an ON response to a step of light. The authors show that pRGC10s primarily stratify in the ON sublamina of the IPL and have a unimodal response to a drifting bar of light. However, a more direct and convincing way to test the polarity of pRGC10s is to measure their response to a full-field step of light. This is a critical experiment for differentiating pRGC10s from ON-OFF and OFF DSGCs that have been identified in other mammals.

A distinguishing feature of the ON-DSGCs was that they showed weak or no response to a full-field, 2 second step of light. We had suspected this was due to surround suppression, which was unavoidable since the DSGCs were generally not in the center of the scan fields and therefore it was necessary to use full-field stimulation. We now show that an ON response can be elicited if cells are stimulated with smaller spots centered on the soma, whereas full-field stimulation suppresses the response (Extended Data Fig. 4A) consistent with surround suppression that has been described previously in ON-DSGCs in other mammals (rabbit: Hoshi et al., 2011; mouse: Goetz et al., 2022 and accompanying dataset on RGCTypes.org). Note that although there was some indication of a small OFF-response to this spot stimulus (Ext. Data Fig. 4A, arrows), no OFF response was observed for drifting bars presented at a range of different velocities (Ext. Data Fig. 4C-D).

The presence of rudimentary anatomical and functional OFF inputs to ON-DSGCs is in line with numerous previous studies of these cells in mouse (Ackert et al., 2009; Goetz et al., 2022; Harris and Dunn, 2023; Sabbah et al., 2017; Sun et al., 2006), further highlighting the conservation of the primate cell type. Moreover, Patterson et al., (2022) showed anatomical OFF bipolar input to the very sparse OFF dendrites in recursive monostriated cells, which we believe are the ON-DSGCs. We highlight similar sparse OFF dendritic processes in Fig. 3B (lower panel) and in Supplemental Video 6. Note that transient ON-OFF responses were readily identified in mouse RGC recordings to drifting bar stimuli (Ext. Data Fig. 5C), whereas similar responses were not seen to drifting bar stimuli in macaque. Overall, the balance of evidence from the morphology, drifting bar stimuli and molecular orthology supports identification of an ON-DSGC rather than an ON-OFF type cell.

o Direction Selectivity: It is difficult to assess whether pRGC10s are indeed direction selective. Figure 2f shows that while they are significantly more direction selective than other RGCs, their DSIs are still quite low compared to previous descriptions of both ON and ON-OFF DSGCs (e.g. Sabbah et al., 2017 who also used calcium imaging). Further, the preferred directions of ON DSGCs are known to cluster into 3 (or maybe 4) directions (Oyster and Barlow, 1967; Sabbah et al., 2017), but Figure 2g shows that there is wide variability in the preferred directions of pRGC10s. The analogous data from mice in Extended Data Figure 4d, e present similar concerns. To overcome these challenges, the authors could first establish a benchmark by demonstrating that they can reliably detect direction selective responses from known populations of DSGCs in mouse. This can be achieved by performing the same calcium imaging experiments on transgenic lines in which ON (*Hoxd10*, *Spig1*) or ON-OFF (*Drd4*, *Trhr*) DSGCs are labeled, or by retrogradely labeling ON DSGCs via central injection into accessory optic nuclei. This experiment would clarify the magnitude of DSI and variance of preferred direction to expect given the authors' experimental paradigm. It would also facilitate comparison to previous work on DSGCs. Second, the authors should consider increasing the number of replicates performed on the pRGC10s in order to solidify the validity of their results given the large variance and small effect size in Figure 2f.

We have included new experimental data to address these concerns. First, to establish a benchmark for DS responses under our experimental conditions, we performed further calcium

imaging experiments followed by molecular identification to show that mouse ON-DSGCs (molecular type Novel_10, Goetz et al., 2022; Huang et al., 2022; Tran et al., 2019) cluster into types with three preferred directions (Ext. Data Fig. 5e-f), consistent with previous studies in mouse and rabbit (Dhande et al., 2013; Goetz et al., 2022; Oyster and Barlow, 1967; Yonehara et al., 2016). By using a higher acquisition frame rate (the same frame rate used for primate recordings), we could distinguish any mouse ON-OFF DSGCs that had previously confounded our interpretation of ON-DSGC preferred directions. Interestingly, we did not find evidence for a fourth mouse ON-DSGC cluster with a preference for temporo-nasal motion on the retina (naso-temporal motion in the visual field) as reported in Sabbah et al., 2017.

With respect to the magnitude of the DSI. The normalized vector sum we obtained for mouse ON-DSGCs (mean±s.d. 0.37 ± 0.16 , median 0.37) was in line with the normalized vector sum values from spike recordings of mouse ON-DSGCs retrogradely labeled from the MTN (median 0.3-0.4, see Fig. 4K of Harris and Dunn, 2023). It is important to note that our approach, similar to that of Harris & Dunn, is unbiased in that all recorded cells that exceeded a signal:noise threshold were included in analysis, regardless of their DSI. This is in contrast to most studies where the DSI itself is used to classify ON-DSGCs as direction-selective (e.g. Sun et al., 2006, Sabbah et al, 2017, Dhande et al., 2013). Overall, the normalized vector sum and preferred directions of the ON-DSGCs we obtained in mouse are comparable to previous reports, which we feel validates our approach and provides a benchmark for the primate recordings.

To address the concern about the number of replicates, we obtained additional recordings from four macaque pRGC10 cells, which are included in the revised Fig. 2f. The normalized vector sum obtained for pRGC10 (0.19 ± 0.11 , mean \pm s.d.), whilst lower than that obtained for mouse ON-DSGCs, was significantly higher than other primate RGCs ($p = 7.45 \times 10^{-6}$, Cohen's effect size - 0.36, medium).

With respect to preferred directions of primate ON-DSGCs, the current sample size, together with the inherent rotational errors associated with tissue alignment, do not allow us to draw definitive conclusions. We acknowledge this limitation in the results. We aim to address this in future experiments in the lab and will also consider the possibility that preferred directions align with optic flow fields (Sabbah et al., 2017), but this would require considerably more data and is beyond the scope of the present study.

o Speed Tuning: ON DSGCs in lower mammals have both direction and speed tuning (Oyster et al., 1972.; Sivyer et al., 2019; Summers and Feller 2022). Together, these characteristics are critical to their role in subserving the optokinetic reflex and differentiate them from other RGC types. The authors should therefore test the speed tuning of pRGC10s. If pRGC10s are tuned to slow stimuli, it would greatly strengthen the claim that they are homologous to ON DSGCs. This would also validate the authors' choice to use glycine receptor subunits to identify pRGC10s in the first place.

We were able to obtain new data from four cells where we examined speed tuning. The results showed some variability, but the overall trend was of strong responses at slow speeds with

attenuation for higher speeds, consistent with findings in other species (Mani et al., 2022; Oyster et al., 1972; Sivyer et al., 2019; Summers and Feller, 2022). These data are included in Ext. Data Fig. 4.

- Mosaic Analyses: The authors claim that pRGC10 somas were randomly distributed across the retina, indicating that this cluster consists of multiple RGC types. However, multiple overlapping mosaics should not make a random spatial distribution. The authors should clarify this point. Further, it would be valuable to know how many independent mosaics exist within the pRGC10 cluster and to relate this to the number of ON DSGCs that have been identified in other species. The number of constituent mosaics in the pRGC10 cluster can be computed by examining finer details of the spatial autocorrelations and density recovery profiles (Cook and Podugolnikova, 2001). This is especially important given the wide variance in preferred direction shown in figure 2f. It would also be valuable to perform mosaic analyses for the BNC2+ human RGCs in order to provide more convincing evidence that these cells are the same as those identified in the macaque.

We thank the reviewer for the excellent suggestion to look at finer details of the spatial autocorrelations. Indeed, we found that the empirical pRGC10 density recovery profile displayed a “well”-like depression consistent with a simulated polymosaic comprising at least three regular mosaics (Cook and Podugolnikova, 2001). This new analysis has been added to Extended Data Fig. 1.

Supporting this finding, we now provide further evidence from fluorescence in situ hybridization experiments that suggests there are multiple cell types within the pRGC10 cluster (Extended Data Fig. 2). The results indicate the presence of a subset of macaque pRGC10 cells that express *Fstl4* (SPIG1). In mouse, *Fstl4* is expressed only in MTN-projecting ON-DSGCs that prefer upward motion in the visual field (Yonehara et al., 2009, 2008). Taken together, these data suggest the existence of molecular distinctions between the cells in the RGC10 cluster and are in general agreement with the mosaic analysis suggesting the presence of multiple cell types. We also note that our calculated coverage factor of 3.5 for the ON-DSGCs aligns with data from the Dacey lab (Kim et al., 2022) showing that the recursive monostratified cells are made up of at least 3 populations, each with a coverage factor of 1.2 and a total coverage factor of 3.5. Notably, the recursive monostratified RGCs described in Kim et al., 2022 were retrogradely labeled from pretectal areas including the NOT-DTN, adding further support to the notion that the cells we have identified are AOS-projecting ON-DSGCs.

With respect to the human data, we performed Voronoi domain analysis for these samples and report those mosaic statistics in the text - the results are similar to those obtained in macaque. The DRP analysis is noisy due to lower cell numbers and unfortunately we have not been able to obtain additional human retinas with sufficiently short post-mortem delay to permit immunostaining for this analysis. We plan to pursue this in follow up studies.

- GABAzine Experiments: The authors assert that their data “indicate that directional selectivity in primate ON-DSGCs depends critically on GABAergic inhibition from SACs.” This claim is unsubstantiated by the data. First, the GABAzine experiment in figure 3 requires further

replicates (current sample size is 2 cells) in order to be properly interpreted. More concerning, however, is that the current data do not suggest the direct involvement of SACs. One would expect that if direction selectivity in pRGC10s relied on SAC input, GABA_A would increase responses to null direction stimuli and leave the responses to preferred direction stimuli relatively unchanged. The data in figure 3g do not confirm this prediction: In the first cell, GABA_A appears to decrease the response to preferred direction stimuli while leaving null direction stimuli relatively unchanged. In the second cell, GABA_A does not appear to abolish direction selectivity, but simply changes the cell's preferred direction. The experiment does not support the authors' intentions to show that the circuitry underlying the direction selectivity is similar to the SAC circuit shown in other species. Otherwise, additional experiments would be required to clarify the contribution of SACs to pRGC10 direction selectivity, such as increasing the sample size and perhaps patching directly from pRGC10s to measure the extent to which inhibitory inputs are direction selective.

We agree with the reviewer that the low sample size for this figure made interpretation difficult and have now obtained 4 additional cells to substantiate this important conclusion (Fig 3, n = 6 cells). Indeed, these recordings show that GABA_A increased responses to null direction stimuli while leaving responses to preferred direction stimuli relatively unchanged, as would be predicted based on prior electrophysiological studies in other species. One of the challenges of performing lengthy pharmacological experiments during imaging is the slow and variable extrusion of the calcium indicator from the cells during the course of the experiment. In one of the earlier recordings, we note that the reduced amplitude in GABA_A may have been due to such extrusion, which would affect measured amplitude of the responses, but not the tuning.

With respect to the apparent reversal of direction selectivity seen in the presence of gabazine in one of the original cells. Similar effects were observed previously in rabbit retina with high concentrations of picrotoxin (Ackert et al., 2009). Again, this anecdotal observation supports the notion that the circuitry is conserved between primate and other mammals. For the newer recordings, we lowered the concentration of gabazine from 20 μ M to 10 μ M and did not observe such reversal.

Minor Points

- In some species, AOS-projecting RGCs are displaced, with their soma in the inner nuclear layer. The authors should comment on whether any fraction of pRGC10s are displaced. We considered this possibility and imaged the INL in all samples used for mosaic analysis. We did not find evidence of any displaced pRGC10s in primate or human retina. We have added a statement to this effect in the results.
- Line 31: "In lower mammals, ON DSGCs are the first site of directional computation...". The first site of direction selectivity is the SAC. This point should be clarified, especially because figure 3 examines direction selective inhibition, presumably from SACs, to ON DSGCs. Thanks. We have made the correction.

- Von Mises fits in figure 2e and extended figure data figure 4c are not very helpful and it is not clear what they add considering the tuning curves of these cells tend to not be very symmetric. The reviewer makes a reasonable point, but we would like to include them for purposes of comparison with previous studies where such fits are shown.

- The authors should report the speed of the drifting bars used on primate tissue in degrees per second and define its contrast and absolute light intensity.
Done.

- Line 150: "These data strongly suggest that a functionally homologous ON-DSGC is present in the human retina." This statement is overly strong and would require substantially more experimentation to justify. Consider removing the word "strongly".
Agreed. Thanks.

Referee #2 (Remarks to the Author):

The manuscript by Wang and Colleagues identifies, for the first time, ON-direction selective retinal ganglion cells (ON DSGCs) in the non-human primate retina and provides compelling evidence for their presence in the human retina.

Background and significance: The retinas of mice, rabbits, and other prominent vertebrate models exhibit many ganglion cells that are direction selective. However, the presence of DSGCs in primates has remained elusive. It has been broadly assumed that DSGCs are absent from the primate retina, and cortical circuits that compute directional signals de novo are the exclusive source of explicit representation of motion direction in the primate brain. At tension with this view were two observations. First, there remain many morphologically defined ganglion cell types whose function is unexamined. Second, the primate retina contains starburst amacrine cells, a critical component in the production of DS signals in the retinas of all 'lower' vertebrates. The conclusive discovery of ON DSGCs in the primate retina, and their genetic, morphological, and physiological similarity to the ON DSGCs in lower vertebrates, is highly significant for our understanding of primate (including human) visual processing and for our understanding of the evolution of vision across vertebrates. In my estimation this is one of the most significant discoveries related to the retina and early visual processing over the last decade.

Data and Methodology: The approach developed here of leveraging transcriptional signatures of ON DSGCs from the mouse to identify candidate ON DSGCs from primate transcriptomics data is ingenious. Overall, the methodological approaches used in this manuscript are all executed to the highest standards and are clearly presented.

Robustness and Reliability: The confluence of gene expression, morphology, circuit connectivity, physiology and pharmacology, provide an exceedingly convincing case that support the discovery of ON DSGCs in the primate retina.

Clarity and Scholarliness. The manuscript is very clearly written. The figures present exactly what a reader wants to know. The prior literature is well cited.

Suggested improvements. This is probably the most complete and solid manuscript for an initial submission that I have had the pleasure of reviewing. It reads like a manuscript that has already undergone a round or two of reviews. Many controls are already present, and I struggle to identify additional experiments or analyses that I would want to see. If I were to nit-pick, it would be very nice if the author could show, or at least comment on whether they think there are 3 or 4 directions (mosaics) of these ON DSGCs. As the authors are no doubt familiar, recent work from the Berson lab indicates the presence of 4 ON DSGC types with 4 different direction preferences, while prior work from Oyster and colleagues in the rabbit indicates just 3. One perspective is that the axes of the ON DGCS are registered to the axes of self-motion (Berson and colleagues), while another perspective is that they are registered to the axis of rotation represented by the semi-circular canals (Simpson and Soodak). But this is only a suggestion for a slight improvement; the study and manuscript are already quite impressive.

We thank the reviewer for their enthusiasm and positive review of this work. We agree that the confluence of methodologies provides a strong case for this discovery. We hope that the revisions that we have made will further strengthen the manuscript.

We agree that resolving the number of mosaics would facilitate comparison to ON-DSGC circuits in other mammals. In an effort to address this question, we have performed additional mosaic and molecular analyses (Extended Data Fig. 1 & 2) that suggest the presence of at least 3 mosaics of ON DSGCs. The calculated coverage factor for pRGC10 of 3.5 lends further support to this conclusion and aligns with the coverage factor obtained for primate recursive monostratified RGCs from Kim et al., 2022. Further functional experiments will be needed to confirm the directional preferences of the primate ON-DSCGs, as the current sample size, together with the inherent rotational errors associated with tissue alignment, precludes definitive conclusions. We acknowledge this limitation in the results.

References

- Ackert JM, Farajian R, Völgyi B, Bloomfield SA (2009) GABA blockade unmask an OFF response in ON direction selective ganglion cells in the mammalian retina. *J Physiol* 587:4481–4495.
- Cook JE, Podugolnikova TA (2001) Evidence for spatial regularity among retinal ganglion cells that project to the accessory optic system in a frog, a reptile, a bird, and a mammal. *Vis Neurosci* 18:289–297.
- Dhande OS, Estevez ME, Quattrochi LE, El-Danaf RN, Nguyen PL, Berson DM, Huberman AD (2013) Genetic dissection of retinal inputs to brainstem nuclei controlling image stabilization. *J Neurosci* 33:17797–17813.
- Distler C, Hoffmann KP (2001) Cortical input to the nucleus of the optic tract and dorsal terminal nucleus (NOT-DTN) in macaques: a retrograde tracing study. *Cereb Cortex* 11:572–580.
- Goetz J, Jessen ZF, Jacobi A, Mani A, Cooler S, Greer D, Kadri S, Segal J, Shekhar K, Sanes JR, Schwartz GW (2022) Unified classification of mouse retinal ganglion cells using function, morphology, and gene expression. *Cell Rep* 40:111040.
- Harris SC, Dunn FA (2023) Asymmetric retinal direction tuning predicts optokinetic eye movements across stimulus conditions. *Elife* 12.
- Huang W, Xu Q, Su J, Tang L, Hao Z-Z, Xu C, Liu R, Shen Y, Sang X, Xu N, Tie X, Miao Z, Liu X, Xu Y, Liu F, Liu Y, Liu S (2022) Linking transcriptomes with morphological and functional phenotypes in mammalian retinal ganglion cells. *Cell Rep* 40:111322.
- Itaya SK, Van Hoesen GW (1983) Retinal axons to the medial terminal nucleus of the accessory optic system in old world monkeys. *Brain Res* 269:361–364.

- Kim YJ, Peterson BB, Crook JD, Joo HR, Wu J, Puller C, Robinson FR, Gamlin PD, Yau K-W, Viana F, Troy JB, Smith RG, Packer OS, Detwiler PB, Dacey DM (2022) Origins of direction selectivity in the primate retina. *Nat Commun* 13:1–20.
- Mani A, Yang X, Zhao T, Leyrer ML, Schreck D, Berson DM (2022) A retinal circuit that vetoes optokinetic responses to fast visual motion. *bioRxiv*.
- Oyster CW, Barlow HB (1967) Direction-selective units in rabbit retina: distribution of preferred directions. *Science* 155:841–842.
- Oyster CW, Takahashi E, Collewijn H (1972) Direction-selective retinal ganglion cells and control of optokinetic nystagmus in the rabbit. *Vision Res* 12:183–93.
- Patterson SS, Bemby BN, Mazzaferri MA, Neitz M, Rieke F, Soetedjo R, Neitz J (2022) Conserved circuits for direction selectivity in the primate retina. *Curr Biol* 32:2529–2538.e4.
- Sabbah S, Gemmer JA, Bhatia-Lin A, Manoff G, Castro G, Siegel JK, Jeffery N, Berson DM (2017) A retinal code for motion along the gravitational and body axes. *Nature* 546:492–497.
- Sivyer B, Tomlinson A, Taylor WR (2019) Simulated Saccadic Stimuli Suppress ON-Type Direction-Selective Retinal Ganglion Cells via Glycinergic Inhibition. *J Neurosci* 39:4312–4322.
- Summers MT, Feller MB (2022) Distinct inhibitory pathways control velocity and directional tuning in the mouse retina. *Curr Biol* 32:2130–2143.e3.
- Sun W, Deng Q, Levick WR, He S (2006) ON direction-selective ganglion cells in the mouse retina. *J Physiol* 576:197–202.
- Telkes I, Distler C, Hoffmann KP (2000) Retinal ganglion cells projecting to the nucleus of the optic tract and the dorsal terminal nucleus of the accessory optic system in macaque monkeys. *Eur J Neurosci* 12:2367–2375.
- Tran NM, Shekhar K, Whitney IE, Jacobi A, Benhar I, Hong G, Yan W, Adiconis X, Arnold ME, Lee JM, Levin JZ, Lin D, Wang C, Lieber CM, Regev A, He Z, Sanes JR (2019) Single-cell profiles of retinal neurons differing in resilience to injury reveal neuroprotective genes. *Neuron* 104:1–17.
- Yonehara K et al. (2016) Congenital Nystagmus Gene FRMD7 Is Necessary for Establishing a Neuronal Circuit Asymmetry for Direction Selectivity. *Neuron* 89:177–193.
- Yonehara K, Ishikane H, Sakuta H, Shintani T, Nakamura-Yonehara K, Kamiji NL, Usui S, Noda M (2009) Identification of retinal ganglion cells and their projections involved in central transmission of information about upward and downward image motion. *PLoS One* 4:e4320.
- Yonehara K, Shintani T, Suzuki R, Sakuta H, Takeuchi Y, Nakamura-Yonehara K, Noda M (2008) Expression of SPIG1 reveals development of a retinal ganglion cell subtype projecting to the medial terminal nucleus in the mouse. *PLoS One* 3:e1533.

Reviewer Reports on the First Revision:

Referees' comments:

Referee #1 (Remarks to the Author):

The authors have thoroughly answered the reviews. The clarification of the ON responses, speed tuning, pharmacology, and mosaic analysis have greatly improved the manuscript and better support the conclusion in naming these ganglion cells "ON direction selective ganglion cells."

The only conclusion that may be stated in a slightly biased, but not incorrect, way is to consistently state that the mosaic analysis suggests "3 or more independent mosaics." The density recovery profile suggests 5 mosaics. The number 3 would be consistent with the original discovery of these neurons in rabbit retina, but the density recovery profile does not identify 3 as a lower limit on the number of independent mosaics. The authors may choose to consider rephrasing how they describe the results of the mosaic analysis.

Referee #2 (Remarks to the Author):

In my view, this is a fantastic study and manuscript that is ready for publication. I would like to congratulate the authors on their thorough responses to the prior round of reviews. I strongly support publication.

While it is extraordinarily unusual for me to do so, I would like to state that I disagree with some of the comments made by reviewer 1.

First, I disagree with the implicit assumption of Reviewer 1 (bullet point 1) that the projection pattern of ON-DSGCs in the primate retina must be the same as that in rodents. The genetic comparison performed by the authors establishes a clear homology that would stand just as strong if the projection patterns differed from that in rodents. Would it be so shocking if RGCs in different species exhibit difference in their projection patterns? Rodents don't exhibit magnocellular layers, but we still think they have a homolog of parasol/alpha RGCs. Thus, I believe the authors have more than adequately addressed this comment by Reviewer 1.

Second, ON-DSGCs in other species typically do not respond well to full-field steps of light (bullet point #2 of reviewer 1). I have experience measuring responses from these cells in mouse and rat. In neither species do responses to full-field flashes cleanly distinguish ON from ON-OFF DSGCs. As the authors note, full-field flashes engage surround suppression and other nonlinearities that make interpretation difficult. Even localized spots in the receptive field center are not particularly reliable. As the authors note, numerous studies have documented OFF input to 'ON-DSGCs'. Furthermore, the relative strength of ON and OFF input to ON-OFF DSGCs is quite variable from cell to cell. These two facts make flashing spots an unreliable tool for segregating ON from ON-OFF DSGCs. The dynamics of response to drifting bars and the preference for lower speeds (both used by the authors) are much more reliable approaches for segregating the ON from ON-OFF DSGCs. Thus, I believe the authors have adequately addressed this comment by Reviewer 1.

Third, I agree with the authors that the values of their DSIs are consistent with previous studies of DSGCs, particularly ON-DSGCs (bullet point #3). I think the control experiments and analyses that the authors have performed in mice adequately address this point.

It was a pleasure to re-read this paper.

Author Rebuttals to First Revision:

Response to reviews

Referees' comments:

Referee #1 (Remarks to the Author):

The authors have thoroughly answered the reviews. The clarification of the ON responses, speed tuning, pharmacology, and mosaic analysis have greatly improved the manuscript and better support the conclusion in naming these ganglion cells "ON direction selective ganglion cells."

The only conclusion that may be stated in a slightly biased, but not incorrect, way is to consistently state that the mosaic analysis suggests "3 or more independent mosaics." The density recovery profile suggests 5 mosaics. The number 3 would be consistent with the original discovery of these neurons in rabbit retina, but the density recovery profile does not identify 3 as a lower limit on the number of independent mosaics. The authors may choose to consider rephrasing how they describe the results of the mosaic analysis.

To address this suggestion, we have revised the main text to state that the mosaic analysis for pRGC10 is consistent with the presence of "multiple" mosaics rather than saying "3 or more independent mosaics". In the figure legend for Extended Data Fig. 1 we state that the density recovery profile analysis suggests up to 5 mosaics. It should be noted that as the number of mosaics increases, the distribution will approach the random distribution thus making it more difficult to accurately resolve a difference between 3, 4 or 5 mosaics. The inclusion of new data in Extended Data Fig. 2 is intended to better address the question as to the number of different populations of ON-DSGCs.

Referee #2 (Remarks to the Author):

In my view, this is a fantastic study and manuscript that is ready for publication. I would like to congratulate the authors on their thorough responses to the prior round of reviews. I strongly support publication.

While it is extraordinarily unusual for me to do so, I would like to state that I disagree with some of the comments made by reviewer 1.

First, I disagree with the implicit assumption of Reviewer 1 (bullet point 1) that the projection pattern of ON-DSGCs in the primate retina must be the same as that in rodents. The genetic comparison performed by the authors establishes a clear homology that would stand just as strong if the projection patterns differed from that in rodents. Would it be so shocking if RGCs in different species exhibit difference in their projection patterns? Rodents don't exhibit magnocellular layers, but we still think they have a homolog of parasol/alpha RGCs. Thus, I believe the authors have more than adequately addressed this comment by Reviewer 1.

Second, ON-DSGCs in other species typically do not respond well to full-field steps of light (bullet point #2 of reviewer 1). I have experience measuring responses from these cells in mouse and rat. In neither species do responses to full-field flashes cleanly distinguish ON from ON-OFF DSGCs. As the authors note, full-field flashes engage surround suppression and other nonlinearities that make interpretation difficult. Even localized spots in the receptive field center are not particularly reliable. As the authors note, numerous studies have documented OFF input to 'ON-DSGCs'. Furthermore, the relative strength of ON and OFF input to ON-OFF DSGCs is quite variable from cell to cell. These two facts make flashing spots an unreliable tool for segregating ON from ON-OFF DSGCs. The dynamics of response to drifting bars and the preference for lower speeds (both used by the authors) are much more reliable

approaches for segregating the ON from ON-OFF DSGCs. Thus, I believe the authors have adequately addressed this comment by Reviewer 1.

Third, I agree with the authors that the values of their DSIs are consistent with previous studies of DSGCs, particularly ON-DSGCs (bullet point #3). I think the control experiments and analyses that the authors have performed in mice adequately address this point.

It was a pleasure to re-read this paper.

We thank the reviewer for their helpful comments.